# Rethinking Generalization in Few-Shot Classification

**Markus Hiller**[*1]    **Rongkai Ma**[*2]    **Mehrtash Harandi**[2]    **Tom Drummond**[1]

[1]School of Computing and Information Systems, The University of Melbourne
[2]Department of Electrical and Computer Systems Engineering, Monash University
markus.hiller@student.unimelb.edu.au
{rongkai.ma, mehrtash.harandi}@monash.edu
tom.drummond@unimelb.edu.au

## Abstract

Single image-level annotations only correctly describe an often small subset of an image's content, particularly when complex real-world scenes are depicted. While this might be acceptable in many classification scenarios, it poses a significant challenge for applications where the set of classes differs significantly between training and test time. In this paper, we take a closer look at the implications in the context of *few-shot learning*. Splitting the input samples into patches and encoding these via the help of Vision Transformers allows us to establish semantic correspondences between local regions across images and independent of their respective class. The most informative patch embeddings for the task at hand are then determined as a function of the support set via online optimization at inference time, additionally providing visual interpretability of '*what matters most*' in the image. We build on recent advances in unsupervised training of networks via masked image modelling to overcome the lack of fine-grained labels and learn the more general statistical structure of the data while avoiding negative image-level annotation influence, *aka* supervision collapse. Experimental results show the competitiveness of our approach, achieving new state-of-the-art results on four popular few-shot classification benchmarks for 5-shot and 1-shot scenarios.

## 1 Introduction

Images depicting real-world scenes are usually comprised of several different entities, e.g., a family walking their dog in a park surrounded by trees, or a person patting their dog (Figure 1). Nevertheless, popular computer vision datasets like ImageNet [39] assign a single *image-level* annotation to classify their entire content. Hence, such a label only correctly applies to an often small subset of the actual image. As a result, models trained on such data via gradient-based methods learn to ignore all seemingly irrelevant information, particularly entities that occur across differently labelled images. While this might be acceptable for conventional classification methods that encounter a diverse number of training examples for all classes they are expected to distinguish, it poses a major but often overlooked challenge for applications where the set of classes differs between training and test time. One such affected area is few-shot learning (FSL) where approaches are expected to correctly classify entirely new classes at test time that have never been encountered during training, just by being provided with a few (e.g., one or five) samples for each of these new categories. During test time, entities that have not been part of the set of training classes and have possibly been perceived as irrelevant might very well be part of the set of test classes – yet, the method was taught to ignore these. Similarly, a method might overemphasize the importance of certain image patterns learned during training that are however of no relevance for the test classes, resulting in *supervision collapse* [9].

---

[*]Joint first authorship

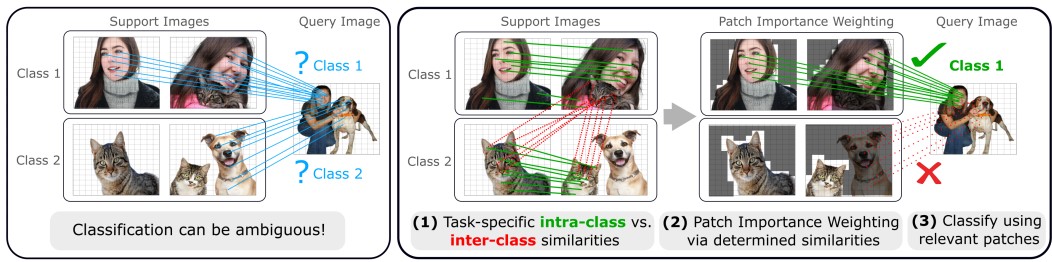

Figure 1: **Tackling classification ambiguity by interpreting images in context.** (Left): Labels assigned to real-world images with multiple entities only correctly describe a subset of the depicted content, leading to ambiguous classification results. (Right): Leveraging intra- and inter-class similarities and differences across the support set allows our method to determine the importance of each individual patch at inference time, i.e., to find out '*what matters most*' in each image. This information is then used to reweight support-query similarities and resolve ambiguity.

Few previous works [9, 19] partially tackle the above challenges. CTX [9] proposes to learn the spatial and semantic alignment between CNN-extracted query and support features using a Transformer-style attention mechanism. The authors further show that self-supervised learning tasks (i.e., SimCLR) can be integrated into episodic training along with normal supervised tasks to learn more generalized features, which benefits solving unseen tasks and mitigates supervision collapse. CAN [19] achieves this in a similar manner by performing cross-attention between class prototypes and query feature maps, highlighting the region of a feature map important for classification during inference. While both methods propose important contributions towards tackling supervision collapse, there exist important drawbacks. Firstly, both methods build their ideas around aligning prototypes based on each query. Such prototypes are merely class-aware and ignore all inter-class information present in the support set – a part that has however been shown to be crucial for few-shot learning [35, 49]. Furthermore, learning query-aligned class representations requires performing the same operation for each query, rendering such approaches rather inefficient at inference time.

Summarizing our and previous works' observations, we aim to address what we see as the two main criteria: 1) building an understanding about an image's structure and content that *generalizes* towards new classes, and 2) providing the ability to interpret the provided samples *in context*, i.e., finding the intra-class similarities and inter-class differences while jointly considering all available information.

**Our work.**[2] To alleviate the negative influence of image-level annotations and to avoid supervision collapse, we decompose the images into patches representing local regions, each having a higher likelihood of being dominated by only one entity. To overcome the lack of such fine-grained annotations we employ self-supervised training with Masked Image Modelling as pretext task [60] and use a Vision Transformer architecture [10] as encoder due to its patch-based nature. We build our classification around the concept of learning task-specific similarity between local regions as a function of the support set at inference time. To this extent, we first create a prior similarity map by establishing semantic patch correspondences between all support set samples irrespective of their class, i.e., also between entities that might not be relevant or potentially even harmful for correct classification (Figure 1, step (1)). Consider the depicted support set with only two classes: 'person' and 'cat'. The lower-right image is part of our support set for 'cat' – and the dog just happens to be in the image. Now in the query sample that shall be classified, the image depicts a person patting their dog. We will thus correctly detect a correspondence of the two dogs across those two images, as well as between the person patches and the other samples of the person support set class. While the correspondences between the person regions are helpful, there is no 'dog' class in the actual support set (i.e., 'dog' is out-of-task information), rendering this correspondence harmful for classification since it would indicate that the query is connected to the image with the 'cat' label. This is where our *token importance weighting* comes into play. We infer an importance weight for each token based on its contribution towards correct classification of the other support set samples, actively strengthening intra-class similarities and inter-class differences by jointly considering all available information – in other words, we learn which tokens 'help' or 'harm' our classification objective (Figure 1, step (2)). These importance-reweighted support set embeddings are then used as basis for our similarity-based query sample classification (step (3)). Our main contributions include the following:

---

[2]Our code is publicly available at `https://github.com/mrkshllr/FewTURE`

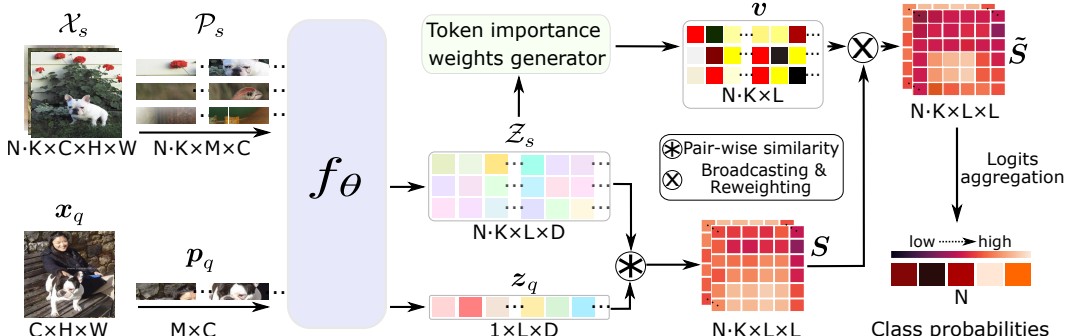

Figure 2: **Illustration of the proposed method FewTURE**. Support and query set images are split into patches and encoded by our Transformer backbone. Classification of query set images is performed by using the reweighted similarity of the encoded patches w.r.t. the support set tokens.

1. We demonstrate that Transformer-only architectures in conjunction with self-supervised pretraining can be successfully used in few-shot settings without the need of convolutional backbones or any additional data.
2. We show that meta fine-tuning of Vision Transformers combined with our inner loop token importance reweighting can successfully use the supervision signal of provided support set labels while avoiding supervision collapse.
3. We provide insights into how establishing general similarities across images independent of classes followed by our optimization-based selection at inference time can boost generalization while allowing visual interpretability at the same time, and show the efficacy of our method by achieving new state-of-the-art results on four popular public benchmarks.

## 2 Few-shot classification via reweighted embedding similarity

We start this section by briefly introducing the problem setting we are tackling in this work: inductive few-shot classification. We then provide an overview of our proposed method *FewTURE* – Few-shot classification with Transformers Using Reweighted Embedding similarity (Figure 2) before elaborating on the main elements in more detail.

**Problem definition**. Inductive $N$-way $K$-shot few-shot classification aims to generalize knowledge learned during training on $\mathcal{D}_{train}$ to unseen test data $\mathcal{D}_{test}$, with classes $\mathcal{C}_{train} \cap \mathcal{C}_{test} = \emptyset$, using only a few labelled samples. We follow the meta-learning protocol of previous works [49] to formulate the few-shot classification problem with episodic training and testing. An episode $\mathcal{E}$ is composed of a support set $\mathcal{X}_s = \{(\boldsymbol{x}_s^{nk}, \boldsymbol{y}_s^{nk}) | n = 1, \dots, N; k = 1, \dots, K; \boldsymbol{y}_s^{nk} \in \mathcal{C}_{train}\}$, where $\boldsymbol{x}_s^{nk}$ denotes the $k$-th sample of class $n$ with label $\boldsymbol{y}_s^{nk}$, and a query set $\mathcal{X}_q = \{(\boldsymbol{x}_q^n, \boldsymbol{y}_q^n) | n = 1, \dots, N\}$, where $\boldsymbol{x}_q^n$ denotes a query sample[3] of class $n$ with label $\boldsymbol{y}_q^n$.

### 2.1 Overview of FewTURE

As depicted in Figure 2, we encode the image patches $\mathcal{P}_s$ of the support set samples along with the query sample patches $\mathbf{p}_q$ via $f_\theta$ and obtain corresponding sets of tokens $\mathcal{Z}_s$ and $\boldsymbol{z}_q$, respectively[4]. It is to be noted that while we choose to illustrate our method via the use of one single query sample, the classification of all query samples is computed at the same time in one single pass in practice. We retrieve our 'prior' correspondence map $\boldsymbol{S}$ expressing the token-wise similarity between the encoded semantic content of the local regions in the query sample and all patches of all support samples, allowing us to consider all available information jointly without incurring information loss due to averaging or similar operations. This *'prior' similarity map* represents correspondences between regions of samples irrespective of their individual class, i.e. also between entities that might not be

---

[3]Without loss of generality, we present our method for the case of one query sample per class to improve ease of understanding. The exact number of query samples per class is generally unknown in practice.

[4]Note that some tensor shapes in the illustrations might differ from the equations for ease of visualization.

relevant or potentially even harmful for correct classification. Using the annotated support set samples, we now infer a task-specific importance weight factor $v^j$ for each support token $z_s^j$ representing its contribution to correctly classify other samples in the support set via online optimization at inference time (Section 2.4). We then reweight the prior similarities to obtain our classification result for the query sample, jointly considering all available information.

## 2.2 Self-supervised pretraining against supervision collapse

To overcome the problem of supervision collapse induced by image-level annotations, we split the input images into smaller parts where each region has a higher likelihood of only containing one major entity and hence a more distinct semantic meaning. Since no labels are available for this more fine-grained data, we encode the information of each local region via an unsupervised method.

We build our approach around the recently introduced idea of using *Masked Image Modeling* (MIM) [3, 60] as a pretext task for self-supervised training of Vision Transformers. In contrast to previous unsupervised approaches [5, 7] which focused mainly on global image-level representations, MIM randomly masks a number of patch embeddings (*tokens*) and aims to reconstruct them given the remaining information of the image. The introduced token constraints help our Transformer backbone to learn an embedding space that yields semantically meaningful representations for each individual image patch. We then leverage the information of the provided labels through fine-tuning the pretrained backbone in conjunction with our inner loop token importance weighting described in the following sections while successfully avoiding supervision collapse (see experimental results in Section 3.2).

## 2.3 Classification through reweighted token similarity

As illustrated in Figure 2, we split each input image $\boldsymbol{x} \in \mathbb{R}^{H \times W \times C}$ into a sequence of $M = {}^{H \cdot W}/{P^2}$ patches $\mathbf{p} = \{p^i\}_{i=1}^M$, with each patch $p^i \in \mathbb{R}^{P^2 \times C}$. We then flatten and pass all patches of the support and query images as input to the Transformer architecture, obtaining the set of support tokens $\mathcal{Z}_s = f_\theta(\mathcal{P}_s)$ with $\mathcal{Z}_s = \{\boldsymbol{z}_s^{nk} | n = 1, \ldots, N, k = 1, \ldots, K\}$, $\boldsymbol{z}_s^{nk} = \{z_s^{nkl} | l = 1, \ldots, L; z_s^{nkl} \in \mathbb{R}^D\}$ and query tokens $\boldsymbol{z}_q = f_\theta(\mathbf{p}_q)$ with $\boldsymbol{z}_q = \{z_q^l | l = 1, \ldots, L; z_q^l \in \mathbb{R}^D\}$. Vision Transformers like ViT [10, 47] satisfy $L = M$ whereas hierarchical Transformers like Swin [28] generally emit a reduced number of tokens $L < M$ due to internal merging strategies.

Having obtained all patch embeddings, we establish semantic correspondences by computing the pair-wise patch similarity matrix between the set of support tokens $\mathcal{Z}_s$ and query tokens $\boldsymbol{z}_q$ as $\boldsymbol{S} \in \mathbb{R}^{N \cdot K \cdot L \times L}$, where each element in $\boldsymbol{S}$ is obtained by $s_{nk}^{l_s, l_q} = \mathrm{sim}(z_s^{nkl_s}, z_q^{l_q})$, where $l_s = 1, \ldots, L$ and $l_q = 1, \ldots, L$. Note that local image regions representing similar entities exhibit higher scores. While any distance metric can be used to compute the similarity ($\mathrm{sim}$), we found cosine similarity to work particularly well for this task. We then use the task-specific token importance weights $\boldsymbol{v} \in \mathbb{R}^{N \cdot K \cdot L \times 1}$ inferred via online optimization based on the annotated support set samples (see Section 2.4) to reweight the similarities through column-wise addition and obtain our task-specific similarity matrix as $\tilde{\boldsymbol{S}} = \boldsymbol{S} + [\boldsymbol{v} \cdot \mathbb{1}^{1 \times L}]$, with elements $\tilde{s}_{nk}^{l_s, l_q}$. Note that this addition of our reweighting logits corresponds to multiplicative reweighting in probability space. We temperature-scale the adapted similarity logits with $1/\tau_S$ and aggregate the token similarity values across all elements belonging to the same support set class via a *LogSumExp* operation, i.e. aggregating $K \cdot L^2$ logits per class followed by a softmax – resulting in the final class prediction $\hat{\boldsymbol{y}}_q$ for the query sample $\boldsymbol{x}_q$ as

$$\hat{\boldsymbol{y}}_q = \mathrm{softmax}\left(\left\{\hat{y}_q^n\right\}_{n=1}^N\right) = \mathrm{softmax}\left(\left\{\log \sum_{k=1}^K \sum_{l_q=1}^L \sum_{l_s=1}^L \exp\left(\tilde{s}_{nk}^{l_s, l_q}/\tau_S\right)\right\}_{n=1}^N\right). \quad (1)$$

## 2.4 Learning token importance at inference time

We use all samples of a task's support set together with their annotations to learn the importance for each individual patch token via online optimization at inference time. As depicted in Figure 3, we formulate the same classification objective as in the previous section but aim to classify the support set samples instead of a query sample. In other words, we use two versions of the support set samples: one with labels as 'support' $\mathcal{Z}_s$ and one without as 'pseudo-query' $\mathcal{Z}_{sq}$, and obtain the similarity

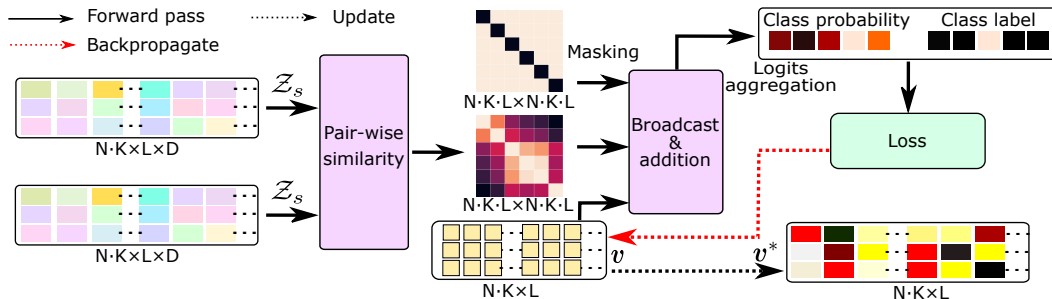

Figure 3: **Inner loop token importance weight generator**. The most helpful tokens for the task at hand are determined as a function of the support set via inner-loop optimization at inference time by reweighting all token similarities based on their contribution towards a correct classification result.

matrix $\boldsymbol{S}_s \in \mathbb{R}^{N \cdot K \cdot L \times N \cdot K \cdot L}$. The token importance weights are initialized to $\boldsymbol{v}^0 = \boldsymbol{0} \in \mathbb{R}^{N \cdot K \cdot L \times 1}$ and column-wise added to form $\tilde{\boldsymbol{S}}_s = \boldsymbol{S}_s + \left[ \boldsymbol{v}^0 \cdot \mathbb{1}^{1 \times N \cdot K \cdot L} \right]$.

The goal is now to determine which tokens of $\mathcal{Z}_s$ are most helpful in contributing towards correctly classifying $\mathcal{Z}_{sq}$, and which ones negatively affect this objective. To prevent tokens from simply classifying themselves, we devise the following strategy for our $N$-way $K$-shot tasks. For scenarios with $K > 1$ samples per class, we apply block-diagonal masking with blocks of size $L \times L$ to the similarity matrix $\tilde{\boldsymbol{S}}_s$ – meaning that we enforce classification of each token in $\mathcal{Z}_{sq}$ exclusively based on information from other images. Since there are no other in-class examples available in 1-shot scenarios, we slightly weaken the constraint and apply local masking in an $m \times m$ window around each patch, forcing the token to be classified based on the remaining information in the image. We found a local window of $m = 5$ to work well throughout our experiments for both architectures.

We use temperature-scaling and aggregate all modified similarity logits across all elements belonging to the same support set class of the annotated $\mathcal{Z}_s$ for each element $\boldsymbol{z}_{sq}^{nk}$, apply a softmax operation and obtain the predicted class probabilities $\hat{\boldsymbol{y}}_s^{nk}$ for each support set sample (*cf.* Equation (1)). Given that the prediction is now dependent on the initialized token similarity weights $\boldsymbol{v}$, we can formulate an online optimization objective by using the support set labels $\boldsymbol{y}_s^{nk}$ as

$$\arg\min_{\boldsymbol{v}} \sum_{n=1}^{N} \sum_{k=1}^{K} \mathcal{L}_{\text{CE}} \left( \boldsymbol{y}_s^{nk}, \ \hat{\boldsymbol{y}}_s^{nk} \left( \boldsymbol{v} \right) \right) . \tag{2}$$

It is to be noted that by using column-wise addition of $\boldsymbol{v}$, we share the importance weights of each support token across all 'pseudo-query' tokens and thus constrain the optimization to jointly learn token importance with respect to all available information. In other words, task-specific intra-class similarities will be strengthened while inter-class ones will be penalized accordingly. We further do not require any second order derivatives like other methods during meta fine-tuning, since the optimization of $\boldsymbol{v}$ is decoupled from the network's parameters.

## 3 Experiments and discussion

### 3.1 Implementation details

Our training strategy is divided into two parts: self-supervised pretraining followed by meta fine-tuning. It is to be noted that each architecture is exclusively trained on the training set data of the respective dataset that is to be evaluated, and no additional data is used.

**Datasets**. We train and evaluate our methods using four popular few-shot classification benchmarks, namely *mini*ImageNet [49], *tiered*ImageNet [38], CIFAR-FS [4] and FC-100 [35].

**Architectures**. We compare two different Transformer architectures in this work, the monolithic ViT architecture [10, 47] in its 'small' form (*ViT-small aka DeiT-small*) and the multi-scale Swin architecture [28] in its 'tiny' version (*Swin-tiny*).

**Self-supervised pretraining**. We employ the strategy proposed by [60] to pretrain our Transformer backbones and mostly stick to the hyperparameter settings reported in their work. Our ViT and Swin

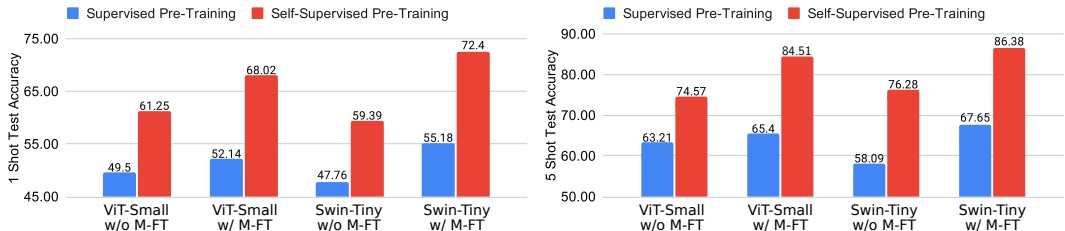

Figure 4: **Supervised vs. self-supervised pretraining.** Average test accuracies on *mini*ImageNet for our method with different pretraining methods, with (*w/*) and without (*w/o*) meta fine-tuning (M-FT).

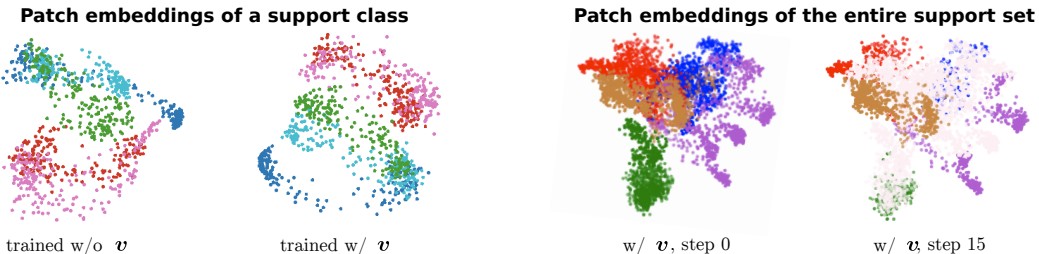

Figure 5: **Instance and class embeddings.** Visualised are the projected tokens of 5 instances of the same novel support set class (left) and of the entire support set (right). From left to right: Instance embeddings meta-trained using our classifier without task-specific token reweighting ('*w/o $v$*') *vs.* trained with 15 reweighting steps ('*w/ $v$*'); Embeddings of entire 5-way 5-shot support set obtained by our approach trained with 15 steps, displayed at reweighting step 0 *vs.* step 15. (PCA projection)

architectures are trained with a batch size of $512$ for $1600$ and $800$ epochs, respectively. We use 4 Nvidia A100 GPUs with 40GB each for our ViT and 8 such GPUs for our Swin models.

**Meta fine-tuning.** We use meta fine-tuning to further refine our embedding space by using the available image-level labels in conjunction with our token-reweighting method. We generally train for up to 200 epochs but find most architectures to converge earlier. We evaluate at each epoch on 600 randomly sampled episodes from the respective validation set to select the best set of parameters. During test time, we randomly sample 600 episodes from the test set to evaluate our model.

**Token importance reweighting and classifier.** We use cosine similarity to compute $S$. While the temperature $\tau_S$ used to scale the logits can be learnt during meta fine-tuning, we found $\tau_S = 1/\sqrt{d}$ to be a good default value (or starting point if learnt, see supplementary material). We use SGD as optimizer with a learning rate of 0.1 for the token importance weight generation.

Please refer to supplementary material of this paper for a more detailed discussion regarding datasets, implementation and hyperparameters.

### 3.2 Self-supervised pretraining and token-reweighted fine-tuning improve generalization

In this section, we investigate the influence of self-supervised pretraining compared to its supervised counterpart using the provided image-level labels (de-facto standard in most current state-of-the-art methods). We only vary the training strategy of our backbone and use our introduced token similarity-based classifier with 15 inner-loop reweighting steps for all experiments. Figure 4 illustrates that both ViT-small and Swin-tiny with self-supervised pretraining alone (*w/o* meta fine-tuning) learn more generalizable features than their respective supervised versions for both 1-shot (left-hand side) and 5-shot (right-hand side) scenarios. In fact, across all cases *w/o* meta fine-tuning, the self-supervised pretrained Transformers outperform their supervised counterparts by more than $10\%$ and up to a significant $18.19\%$ in the case of Swin-tiny. This clearly demonstrates that our self-supervised pretraining strategy explores information that is richer and beyond the labels. Another interesting insight is that the meta fine-tuning stage does not improve supervised backbones as much as their unsupervised counterparts. Specifically after unsupervised pretraining, meta fine-tuning of the Transformers in conjunction with our token-reweighting strategy is able to boost the performance by

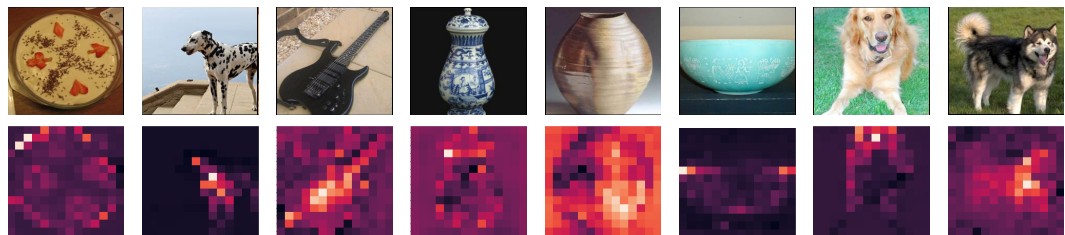

Figure 6: **Learning token importance at inference time.** Visualized importance weights learnt via online optimization for support set samples in a 5-way 5-shot task on the *mini*ImageNet test set.

$6.77\%$ (ViT-small) and $13.01\%$ (Swin-tiny) for 1-shot, and $9.94\%$ (ViT-small) and $10.10\%$ (Swin-tiny) for 5-shot. While such significant improvements cannot be observed across the supervised networks, the Swin versions seem to generally start off lower after pretraining but benefit more from the fine-tuning than ViT. The observed results clearly indicate that our token-reweighted fine-tuning strategy is able to further improve the generalization of self-supervised Transformers, thus performing better on the novel tasks of the unseen test set. Figure 5 additionally depicts projected views of the tokens of 5 instances from a novel class as well as the entire novel support set in embedding space. Representations obtained with our classifier seem to retain the instance information ('*w/o v*') and separation is improved when using token importance reweighting ('*w/ v*'). While the projected tokens of the entire support set show partial overlap between classes as is expected due to commonalities like e.g. similar background, our reweighting clearly determines the class-characteristic tokens (displayed in their original class-respective color). These results indicate that our similarity-based classifier coupled with task-specific token reweighting is able to better disentangle the embeddings of different instances from the same class as well as other classes, which prevents the network from supervision collapse and achieves the higher performance observed on the benchmarks. They further show that self-supervised pretraining is helpful but not sufficient to achieve well-separated representations without supervision collapse that are suitable for few-shot classification.

### 3.3 Selecting helpful patches at inference time

Figure 6 shows a visualization of the patch importance weights $v$ that are learned at inference time during the inner loop adaptation for the support set images. Brighter regions represent a higher importance weight, meaning that these patches will contribute most to the classification of query samples if matches with high similarity can be found. Judging from the visualized weights, *FewTURE* seems to consistently select characteristic regions of the depicted objects, e.g., the rim of the bowls, strings of the guitar or the dogs' facial area, and to exclude unimportant or out-of-task information.

We further investigate the influence of the number of optimization steps in our inner loop token importance weighting method using ViT-small on *mini*ImageNet. The results in Figure 7 indicate that increasing the steps up to 20 aligns with increased performance, both during validation and testing. While the initial increase in test accuracy when using our token reweighting (steps $> 0$) is rather significant with $1.15\%$, the contribution of higher step numbers comes at the cost of higher

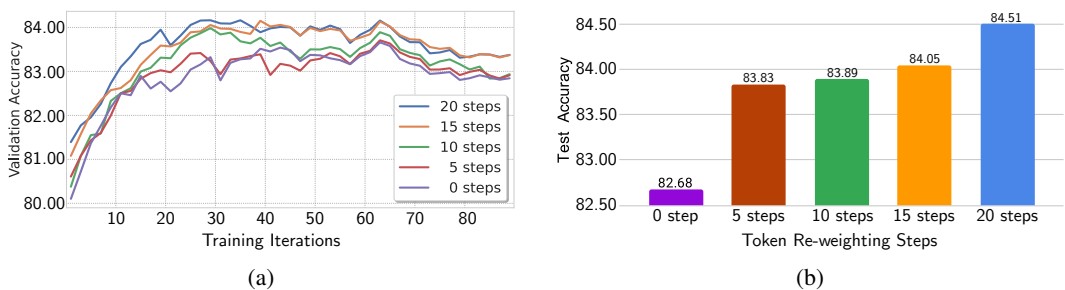

Figure 7: **Inner loop token reweighting.** Average classification accuracies on the *mini*ImageNet validation set (a), and test set (b) for varying inner loop optimization steps, evaluated with a ViT-small backbone and SGD with 0.1 as learning rate.

Table 1: Average classification accuracy for 5-way 1-shot and 5-way 5-shot scenarios. Reported are the mean and 95% confidence interval on the unseen test sets of *mini*ImageNet [49] and *tiered*ImageNet [38], using the established evaluation protocols.

| Model | Backbone | ≈ # Params | *mini*ImageNet | | *tiered*ImageNet | |
|---|---|---|---|---|---|---|
| | | | 1-shot | 5-shot | 1-shot | 5-shot |
| ProtoNet [42] | ResNet-12 | 12.4 M | $62.29_{\pm0.33}$ | $79.46_{\pm0.48}$ | $68.25_{\pm0.23}$ | $84.01_{\pm0.56}$ |
| FEAT [54] | ResNet-12 | 12.4 M | $66.78_{\pm0.20}$ | $82.05_{\pm0.14}$ | $70.80_{\pm0.23}$ | $84.79_{\pm0.16}$ |
| DeepEMD [55] | ResNet-12 | 12.4 M | $65.91_{\pm0.82}$ | $82.41_{\pm0.56}$ | $71.16_{\pm0.87}$ | $86.03_{\pm0.58}$ |
| IEPT [57] | ResNet-12 | 12.4 M | $67.05_{\pm0.44}$ | $82.90_{\pm0.30}$ | $72.24_{\pm0.50}$ | $86.73_{\pm0.34}$ |
| MELR [12] | ResNet-12 | 12.4 M | $67.40_{\pm0.43}$ | $83.40_{\pm0.28}$ | $72.14_{\pm0.51}$ | $87.01_{\pm0.35}$ |
| FRN [50] | ResNet-12 | 12.4 M | $66.45_{\pm0.19}$ | $82.83_{\pm0.13}$ | $72.06_{\pm0.22}$ | $86.89_{\pm0.14}$ |
| CG [59] | ResNet-12 | 12.4 M | $67.02_{\pm0.20}$ | $82.32_{\pm0.14}$ | $71.66_{\pm0.23}$ | $85.50_{\pm0.15}$ |
| DMF [53] | ResNet-12 | 12.4 M | $67.76_{\pm0.46}$ | $82.71_{\pm0.31}$ | $71.89_{\pm0.52}$ | $85.96_{\pm0.35}$ |
| InfoPatch [26] | ResNet-12 | 12.4 M | $67.67_{\pm0.45}$ | $82.44_{\pm0.31}$ | - | - |
| BML [61] | ResNet-12 | 12.4 M | $67.04_{\pm0.63}$ | $83.63_{\pm0.29}$ | $68.99_{\pm0.50}$ | $85.49_{\pm0.34}$ |
| CNL [59] | ResNet-12 | 12.4 M | $67.96_{\pm0.98}$ | $83.36_{\pm0.51}$ | $73.42_{\pm0.95}$ | $87.72_{\pm0.75}$ |
| Meta-NVG [56] | ResNet-12 | 12.4 M | $67.14_{\pm0.80}$ | $83.82_{\pm0.51}$ | $74.58_{\pm0.88}$ | $86.73_{\pm0.61}$ |
| PAL [30] | ResNet-12 | 12.4 M | $69.37_{\pm0.64}$ | $84.40_{\pm0.44}$ | $72.25_{\pm0.72}$ | $86.95_{\pm0.47}$ |
| COSOC [29] | ResNet-12 | 12.4 M | $69.28_{\pm0.49}$ | $85.16_{\pm0.42}$ | $73.57_{\pm0.43}$ | $87.57_{\pm0.10}$ |
| Meta DeepBDC [52] | ResNet-12 | 12.4 M | $67.34_{\pm0.43}$ | $84.46_{\pm0.28}$ | $72.34_{\pm0.49}$ | $87.31_{\pm0.32}$ |
| LEO [40] | WRN-28-10 | 36.5 M | $61.76_{\pm0.08}$ | $77.59_{\pm0.12}$ | $66.33_{\pm0.05}$ | $81.44_{\pm0.09}$ |
| CC+rot [15] | WRN-28-10 | 36.5 M | $62.93_{\pm0.45}$ | $79.87_{\pm0.33}$ | $70.53_{\pm0.51}$ | $84.98_{\pm0.36}$ |
| FEAT [54] | WRN-28-10 | 36.5 M | $65.10_{\pm0.20}$ | $81.11_{\pm0.14}$ | $70.41_{\pm0.23}$ | $84.38_{\pm0.16}$ |
| PSST [8] | WRN-28-10 | 36.5 M | $64.16_{\pm0.44}$ | $80.64_{\pm0.32}$ | - | - |
| MetaQDA [58] | WRN-28-10 | 36.5 M | $67.83_{\pm0.64}$ | $84.28_{\pm0.69}$ | $74.33_{\pm0.65}$ | $89.56_{\pm0.79}$ |
| OM [37] | WRN-28-10 | 36.5 M | $66.78_{\pm0.30}$ | $85.29_{\pm0.41}$ | $71.54_{\pm0.29}$ | $87.79_{\pm0.46}$ |
| FewTURE (ours) | ViT-Small | 22 M | $68.02_{\pm0.88}$ | $84.51_{\pm0.53}$ | $72.96_{\pm0.92}$ | $86.43_{\pm0.67}$ |
| FewTURE (ours) | Swin-Tiny | 29 M | $\mathbf{72.40_{\pm0.78}}$ | $\mathbf{86.38_{\pm0.49}}$ | $\mathbf{76.32_{\pm0.87}}$ | $\mathbf{89.96_{\pm0.55}}$ |

Table 2: Average classification accuracy for 5-way 1-shot and 5-way 5-shot scenarios. Reported are the mean and 95% confidence interval on the unseen test sets of CIFAR-FS [4] and FC-100 [35], using the established evaluation protocols.

| Model | Backbone | ≈ # Params | CIFAR-FS | | FC100 | |
|---|---|---|---|---|---|---|
| | | | 1-shot | 5-shot | 1-shot | 5-shot |
| ProtoNet [42] | ResNet-12 | 12.4 M | - | - | $41.54_{\pm0.76}$ | $57.08_{\pm0.76}$ |
| MetaOpt [22] | ResNet-12 | 12.4 M | $72.00_{\pm0.70}$ | $84.20_{\pm0.50}$ | $41.10_{\pm0.60}$ | $55.50_{\pm0.60}$ |
| MABAS [21] | ResNet-12 | 12.4 M | $73.51_{\pm0.92}$ | $85.65_{\pm0.65}$ | $42.31_{\pm0.75}$ | $58.16_{\pm0.78}$ |
| RFS [46] | ResNet-12 | 12.4 M | $73.90_{\pm0.80}$ | $86.90_{\pm0.50}$ | $44.60_{\pm0.70}$ | $60.90_{\pm0.60}$ |
| BML [61] | ResNet-12 | 12.4 M | $73.45_{\pm0.47}$ | $88.04_{\pm0.33}$ | - | - |
| CG [14] | ResNet-12 | 12.4 M | $73.00_{\pm0.70}$ | $85.80_{\pm0.50}$ | - | - |
| Meta-NVG [56] | ResNet-12 | 12.4 M | $74.63_{\pm0.91}$ | $86.45_{\pm0.59}$ | $46.40_{\pm0.81}$ | $61.33_{\pm0.71}$ |
| RENet [20] | ResNet-12 | 12.4 M | $74.51_{\pm0.46}$ | $86.60_{\pm0.32}$ | - | - |
| TPMN [51] | ResNet-12 | 12.4 M | $75.50_{\pm0.90}$ | $87.20_{\pm0.60}$ | $46.93_{\pm0.71}$ | $63.26_{\pm0.74}$ |
| MixFSL [1] | ResNet-12 | 12.4 M | - | - | $44.89_{\pm0.63}$ | $60.70_{\pm0.60}$ |
| CC+rot [15] | WRN-28-10 | 36.5 M | $73.62_{\pm0.31}$ | $86.05_{\pm0.22}$ | - | - |
| PSST [8] | WRN-28-10 | 36.5 M | $77.02_{\pm0.38}$ | $88.45_{\pm0.35}$ | - | - |
| Meta-QDA [58] | WRN-28-10 | 36.5 M | $75.83_{\pm0.88}$ | $88.79_{\pm0.75}$ | - | - |
| FewTURE (ours) | ViT-Small | 22 M | $76.10_{\pm0.88}$ | $86.14_{\pm0.64}$ | $46.20_{\pm0.79}$ | $63.14_{\pm0.73}$ |
| FewTURE (ours) | Swin-Tiny | 29 M | $\mathbf{77.76_{\pm0.81}}$ | $\mathbf{88.90_{\pm0.59}}$ | $\mathbf{47.68_{\pm0.78}}$ | $\mathbf{63.81_{\pm0.75}}$ |

computational complexity, and we generally found anything between 5 and 15 steps to be a good performance *vs.* inference-time trade-off (see supplementary material for further details).

### 3.4 Evaluation on few-shot classification benchmarks

We conduct experiments using the few-shot settings established in the community, namely 5-way 1-shot and 5-way 5-shot – meaning the network has to distinguish samples from 5 novel classes based on a provided number of 1 or 5 images per class. We evaluate our method *FewTURE* using two different Transformer backbones and compare our results against the current state of the art in Table 1 for the *mini*ImageNet and *tiered*ImageNet, and in Table 2 for the CIFAR-FS and FC100 datasets. It is to be noted that in contrast to previous works, we do not employ the help of any convolutional

backbone but instead (and as far as we are aware for the first time) use a Transformer backbone together with our previously introduced token importance reweighting method to achieve these results. We are able to set new state of the art results across all four datasets in both 5-shot and 1-shot settings, improving particularly the 1-shot results on *mini*ImageNet and *tiered*ImageNet by significant margins of 3.03% and 1.74%, respectively.

### 3.5 Increasing efficiency by pruning the token sequence

To further improve the computational efficiency of our method, we investigate its behaviour when limiting the number of tokens that are considered to establish patch-wise correspondences to only a subset – allowing *FewTURE* to scale to potentially large *many-way many-shot* settings. We use the attention maps inherent in our approach (averaged over all heads) to prune the number of patch tokens by only using the ones within the top-k attention values to compute the similarity matrix $S$. The results obtained with our ViT-small backbone for pruning the number of tokens to 75%, 50%, 25% and 10% of the original token number indicate that such pruning might be an interesting avenue for future work, with our method still achieving 96.4% of its original performance when only retaining 10% of the number of tokens.

Table 3: Pruning the number of tokens. Test accuracy for 5-way 5-shot on *mini*ImageNet [49].

| # tokens | Test Acc. |
|---|---|
| 100% | $84.05 \pm 0.53$ |
| 75% | $83.15 \pm 0.57$ |
| 50% | $83.81 \pm 0.59$ |
| 25% | $81.79 \pm 0.57$ |
| 10% | $81.05 \pm 0.62$ |

### 3.6 Ablation on the type of classifier

Table 4: Changing the classifier. Test accuracy for 5-way 5-shot on *mini*ImageNet [49].

| Classifier | Test Acc. |
|---|---|
| Prototyp. w/ Euclid. Dist. | $82.80 \pm 0.59$ |
| Prototyp. w/ Cosine. Dist. | $79.90 \pm 0.65$ |
| Linear (optimized online) | $82.37 \pm 0.57$ |
| FewTURE ( 0 rew. steps) | $82.68 \pm 0.55$ |
| FewTURE (15 rew. steps) | $\mathbf{84.05 \pm 0.53}$ |

We train a linear classifier as well as prototypical approach using our pre-trained ViT-small backbone for a 5-way 5-shot setting on *mini*Imagenet [49] to investigate the influence of the choice of classifier (Table 4). We were able to obtain a test accuracy of 82.80% for the prototypical network after optimising the pre-trained backbone with meta-finetuning, which is competitive to the results we obtain for our method without reweighting ('0 step' in Figure 7(b)), but is clearly outperformed by our reweighting-based approach. To provide a fair comparison, we optimize the linear classifier at inference time to adapt to the support set and obtained a maximum test accuracy of 82.37%. Both results indicate the quality of embedding our backbone is able to produce but also demonstrate the importance of our task-specific reweighting-based approach. For further ablation studies, please refer to the supplementary material.

### 3.7 Limitations

Our introduced method is arguably more powerful for cases where multiple examples of the same class are provided, i.e. in $N$-way $K$-shot settings with $K > 1$. While FewTURE works well in 1-shot settings (Table 1 and 2), the inner loop adaptation procedure still aims to exclude cross-class similarities that hurt classification performance, but has less diverse information for selecting the most helpful regions due to the lack of other in-class comparison samples – which might thus yield slightly less-refined token selections compared to multi-shot scenarios (*supplementary material*).

While we did not face significant problems regarding the comparably very small size of our datasets (e.g. *mini*ImageNet with 38K compared to the usually used ImageNet with 1.28M training images), more specialized applications with highly limited training data might be negatively impacted and successfully training our method due to the reduced inductive bias present in the Transformer architecture could prove challenging.

## 4 Related work

Over the past few years, the family of few-shot learning (FSL) has grown diverse and broad. Those closely related to this work can be categorized into two groups: metric-based methods [31, 32, 41, 42,

49, 54, 55] and optimisation-based methods [13, 18, 22, 34, 40, 62]. Metric-based methods, such as ProtoNet [42], DeepEMD [55], and RelationNet [44] aim to learn a class representation (prototype) by averaging the embeddings belonging to the same class and employ a predefined (ProtoNet and DeepEMD) or learned (RelationNet) metric to perform prototype-query matching. FEAT [54] and TDM [23] take this a step further and use attention mechanisms to adapt the extracted features to the novel tasks. Our method instead fully utilizes the embeddings of local image regions (patch tokens), preventing loss of information and supervision collapse occurring in the aforementioned prototype-based approaches (see supplementary material).

Optimisation-based methods such as MAML [13] and Reptile [34] propose to learn a set of initial model parameters that can quickly adapt to a novel task. However, updating all model parameters is often not feasible given large backbones and only few labeled samples during inference. To alleviate this so-called meta-overfitting problem, CAVIA [62] and LEO [40] propose to learn and adapt a lower dimensional representation that is mapped onto the network. Our method is inspired by such lower-dimensional adaptation strategies and learns a tiny set of context-aware re-weighting factors online for each novel task without requiring higher-order gradients, resulting in a flexible and efficient framework.

**Self-supervised learning for FSL**. Although self-supervised learning is underrepresented in the context of few-shot learning, some recent works [15, 33, 43] have shown that self-supervision via pretext tasks can be beneficial when integrated as auxiliary loss. S2M2 [33] employs rotation [16] and exemplars [11] along with common supervised learning during the pre-training stage. CTX [9] demonstrates that SimCLR [6] can be combined with supervised-learning tasks in an episodic-training manner to learn a more generalized model. In contrast, FewTURE demonstrates that the self-supervised pretext task (i.e., Masked Image Modelling [2, 3, 17, 25, 36, 45, 60]) can be used in a standalone manner to learn more generalized features for few-shot learning on small-scale datasets.

**Vision Transformers in FSL**. Transformers have been immensely successful in the field of Natural Language Processing. Recent studies [10, 24, 28, 48] suggest that encoding the long-range dependency of data via self-attention also yields promising results for vision tasks (e.g., image classification, joint vision-language modeling, etc.). However, there is an important trade-off between leveraging the rich representation capacity and the lack of inductive bias. Transformers have gained a reputation to generally require significantly more training data compared to convolutional neural networks (CNNs), since properties like translation invariance, locality and hierarchical structure of visual data have to be inferred from the data [27]. While this data-hungry nature largely prevented the use of Transformers in problems with scarce data like few-shot learning, some recent works demonstrate that a single Transformer head can be successfully adopted to perform feature adaptation [9, 54]. To the best of our knowledge, *FewTURE* is the first approach that uses a fully Transformer-based architecture to obtain representative embeddings while being trained exclusively on training data of the respective few-shot dataset.

## 5   Conclusion

In this paper, we presented a novel approach to tackle the challenge of supervision collapse introduced by one-hot image-level labels in few-shot learning. We split the input images into smaller parts with higher probability of being dominated by only one entity and encode these local regions using a Vision Transformer architecture pretrained via Masked Image Modeling (MIM) in a self-supervised way to learn a representative embedding space beyond the pure label information. We devise a classifier based on patch embedding similarities and propose a token importance reweighting mechanism to refine the contribution of each local patch towards the overall classification result based around intra-class similarities and inter-class differences as a function of the support set information. Our obtained results demonstrate that our proposed method alleviates the problem of supervision collapse by learning more generalized features, achieving new state-of-the-art results on four popular few-shot classification datasets.

**Acknowledgements.** The authors would like to thank Zhou *et al.* [60] for sharing their insights and code regarding self-supervised pretraining, as well as Dosovitskiy *et al.* [10], Touvron *et al.* [47] and Liu *et al.* [28] for sharing details of the ViT and Swin architectures.

Parts of this research were undertaken using the LIEF HPC-GPGPU Facility hosted at the University of Melbourne. This Facility was established with the assistance of LIEF Grant LE170100200.

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
