# Rethinking Generalization in Few-Shot Classification

## Supplementary Material

**Markus Hiller**[*1]     **Rongkai Ma**[*2]     **Mehrtash Harandi**[2]     **Tom Drummond**[1]

[1]School of Computing and Information Systems, The University of Melbourne
[2]Department of Electrical and Computer Systems Engineering, Monash University
`markus.hiller@student.unimelb.edu.au`
`{rongkai.ma, mehrtash.harandi}@monash.edu`
`tom.drummond@unimelb.edu.au`

## A    Selecting helpful patches at inference time in 1-shot scenarios

Figure 6 in the main paper demonstrates that our approach is able to successfully learn at inference time which image regions should be considered to classify the unknown query images in a 5-way 5-shot scenario. We additionally present the visualization of the token importance weights for the query images of a 5-way 1-shot scenario in Figure A1. It can be clearly observed that the brighter regions representing higher importance of the respective image patches strongly relate to the actual objects that are to be classified, even in the case of smaller objects (2nd and 4th from the right). While our method only has access to significantly less information in the here presented 1-shot than in the case of 5-shot scenarios (see details in Section 2.4), our proposed way of masking the neighborhood of each pixel during the online optimization procedure still enables selection of the most helpful areas characteristic for the respective classes.

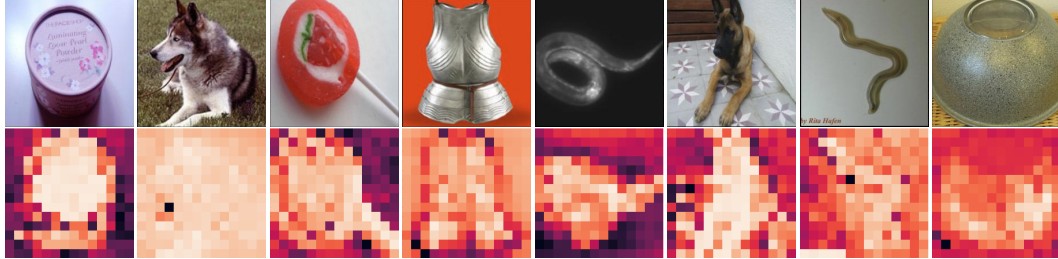

Figure A1: **Learning token importance at inference time.** Visualized importance weights learnt via online optimization for support set samples in a 5-way 1-shot task on the *mini*ImageNet test set.

## B    Discussion on model size and performance

Related works have shown that model size seems to not be a good indicator for few-shot performance, most likely since training datasets are comparably small (e.g. 38.4K images in *mini*ImageNet [20] vs. standard ImageNet with 1.28M [16]) and big networks are thus much more prone to overfit. Chen *et al.* [2] demonstrate in Figure 3 of their paper that the performance gains due to larger backbones plateau across all methods for backbones bigger than ResNet10 in their experiments and only offer diminishing gains (if any at all). The investigations of Mangla *et al.* [11] yielded similar results,

---

[*]Joint first authorship

where the performance on the *mini*ImageNet and *tiered*ImageNet datasets even decreased by around 0.5-1% when scaling up from ResNet18 to ResNet34 (Table 2). We thus conclude that increased number of parameters on its own does not lead to better few-shot performance, and the tendency of many recent works to choose the established ResNet12 (12.4M) over bigger backbones is highly likely a result of this.

To gauge the influence of model size in FewTURE, we additionally investigate the use of the significantly smaller ViT-tiny architecture with only 5M parameters [19]. Results in Table A1 show that our method achieves a competitive accuracy of 81.10% on the *mini*ImageNet test dataset with less than one seventh of the number of parameters of a WRN-28-10, but is (in contrast to many other methods like e.g. [22]) able to leverage increased model sizes to further boost performance.

Table A1: Investigating model size and performance. Average classification accuracy on the *mini*-ImageNet test set, evaluated in a 5-way 5-shot scenario with a ViT-small backbone.

| Method | Backbone | #Params | Test Accuracy |
|--------|----------|---------|---------------|
| ProtoNet [18] | ResNet-12 | $\approx 12.4M$ | $79.46_{\pm0.48}$ |
| FEAT [22] | ResNet-12 | $\approx 12.4M$ | $82.05_{\pm0.14}$ |
| DeepEMD [23] | ResNet-12 | $\approx 12.4M$ | $82.41_{\pm0.56}$ |
| COSOC [10] | ResNet-12 | $\approx 12.4M$ | $85.16_{\pm0.42}$ |
| Meta DeepBDC [21] | ResNet-12 | $\approx 12.4M$ | $84.46_{\pm0.28}$ |
| LEO [17] | WRN-28-10 | $\approx 36.5M$ | $77.59_{\pm0.12}$ |
| CC+rot [7] | WRN-28-10 | $\approx 36.5M$ | $79.87_{\pm0.33}$ |
| FEAT [22] | WRN-28-10 | $\approx 36.5M$ | $81.11_{\pm0.14}$ |
| PSST [4] | WRN-28-10 | $\approx 36.5M$ | $80.64_{\pm0.32}$ |
| MetaQDA [24] | WRN-28-10 | $\approx 36.5M$ | $84.28_{\pm0.69}$ |
| OM [13] | WRN-28-10 | $\approx 36.5M$ | $85.29_{\pm0.41}$ |
| FewTURE (ours) | ViT-Tiny | $\approx 5.0M$ | $81.10_{\pm0.61}$ |
| FewTURE (ours) | ViT-Small | $\approx 22.0M$ | $84.51_{\pm0.53}$ |
| FewTURE (ours) | Swin-Tiny | $\approx 29.0M$ | $\mathbf{86.38}_{\pm0.49}$ |

## C  Discussion on self-supervised vs. supervised pretraining

**Performance in few-shot learning.** We demonstrate in Figure 4 of the main paper that self-supervised pretraining with masked image modelling as pretext task provides a significant advantage over supervised pretraining for our approach – a finding that differs from prior non-few-shot literature where self-supervised methods only moderately outperform their supervised counterparts [25] or even perform worse in some cases [3]. We provide our interpretation and insights regarding this in the following.

Few-shot classification is distinctively different from 'conventional' classification (like investigated in [3]) in one important aspect: novel previously unseen classes are encountered at test time. As such, supervised learning induces a tendency of the representation space to overfit to the structure of the classes observed during training. In other words, the representation space is created and condensed to easily separate observed training classes, but at the expense of distorting other dimensions that might be crucial to correctly distinguish yet unseen classes. This is known in the few-shot literature as '*supervision collapse*' [5]. Since no class labels are provided during the self-supervised pretraining, we expect the method to create a more general/less distorted representation space that is significantly better suited to generalize to yet unseen classes and avoid collapse. These intuitions are supported by the results we have obtained (Fig 4.). We further observe that self-supervised training is helpful to prevent early overfitting when learning from small few-shot datasets (e.g. 38.4K *mini*ImageNet [20] vs. 1.2M ImageNet1K [16]).

**Training details of supervised pretraining.** For adequate comparison to related work in few-shot learning, we follow the widely adopted pretraining scheme used in FEAT [22] and other works (e.g.

DeepEMD [23]) for our supervised pretraining. In detail, we train the network with a cross-entropy loss on the training set of the respective dataset to solve a standard classification task (e.g. for *mini*ImageNet: 64 classes) – i.e., using the exact same data we use for self-supervised pretraining. Like [22] we use the representations of the penultimate layer (before the classifier) to evaluate the performance and quality of the embeddings. To judge suitability of the encoder for few-shot tasks, an N-way 1-shot task is commonly solved (e.g. N=16 for *mini*ImageNet due to the 16 classes in the validation set) – and we tried three different variants here:

1. & 2. One sample per class is encoded to produce a class-embedding ('prototype'), and classification performance is evaluated using 15 queries per class (as used in recent related works). To retrieve one embedding per sample, we use the average over all patch tokens produced by the Transformer architecture. For fairness regarding metrics, we evaluate both:
   1. embedding distance (MSE) and
   2. embedding similarity (cosine) to perform classification.
3. We additionally use our own patch-based classifier to evaluate the few-shot setting using all patch embeddings (as we later do during fine-tuning & evaluation).

We perform validation over 200 such few-shot tasks after every epoch during training and pick the best-performing model regarding highest average validation accuracy. We encountered clear signs of overfitting during this type of supervised training, with the training accuracy consistently improving to convergence, but validation accuracy plateauing (or decreasing) rather early on ($\sim$350-500ep), independent of the variant we used to evaluate on the validation set.

## D  Ablation studies on components of FewTURE

In this section, we provide further insights into our approach and the design choices we made.

### D.1  Ablation on inner loop token reweighting

A more detailed version of the average classification test accuracies achieved with a meta fine-tuned ViT backbone on the *mini*ImageNet dataset used for the visualization of the contribution for different numbers of token reweighting steps during online optimisation (main paper, Figure 7) is presented in Table A2, including the respective 95% confidence intervals. As discussed in the main paper, we observed a strong initial increase of 1.15% when using our proposed adaptation via online optimization (steps$> 0$). While a higher number of inner-loop updates seems to still lead to increased accuracy across all our test runs, this benefit brings along higher computational cost as can be seen in the second row of Table A2. We generally found settings between 5 and 15 steps to be a good accuracy *vs.* inference-time trade-off. Our experiments were conducted using an Nvidia-2080ti GPU and the stated inferences times have been averaged over 1800 query sample classifications. It is to be noted that the code has not been specifically optimized for fast inference times, and these values should rather be interpreted in a relative manner.

Table A2: Average classification accuracy and inference times on the *mini*-ImageNet test set for varying inner loop optimization steps, evaluated in a 5-way 5-shot scenario with a ViT-small backbone and SDG with 0.1 as learning rate. Experiments were conducted using an Nvidia-2080ti and runtimes were averaged over 1800 query sample classifications.

|  | 0 steps | 5 steps | 10 steps | 15 steps | 20 steps |
|---|---|---|---|---|---|
| **Accuracy** | $82.68_{\pm0.59}$ | $83.83_{\pm0.59}$ | $83.89_{\pm0.57}$ | $84.05_{\pm0.55}$ | $84.51_{\pm0.53}$ |
| **Inference time** [ms] | $156.86_{\pm2.16}$ | $159.86_{\pm2.12}$ | $162.11_{\pm2.11}$ | $165.62_{\pm2.06}$ | $168.62_{\pm2.22}$ |

### D.2  Ablation on token aggregation and similarity metrics

As discussed in the main paper, we use the *logsumexp* operation to aggregate our similarity logits as it poses a rigorous and numerically stable way of combining individual class probabilities (one for each token) to a valid overall probability distribution over classes for each image, independent of how the individual token (log) probability scores are obtained. Table A3 (a) shows the results of additional

experiments (training and testing) using our method (ViT-small) and 15 token reweighting steps with the only change being aggregation of the logtis via *mean*, and we found it to underperform our chosen *logsumexp* method of aggregation. Direct addition without normalization (i.e. just summing up all logits) proved unstable due to large logit values and was thus not included in this table.

We further investigated the use of alternate metrics to compute the similarity between different tokens. Both the use of the negative Euclidean distance and unscaled dot-product yielded inferior results compared to the temperature-scaled cosine distance we use in FewTURE (Table A3 (b)).

Table A3: Ablation on token aggregation method and similarity metric. Reported are the average classification accuracies on the *mini*ImageNet test set evaluated in a 5-way 5-shot scenario with a ViT-small backbone.

(a) Token aggregation

| Aggregation method | Test Accuracy |
|---|---|
| logsumexp | $\mathbf{84.05 \pm 0.53}$ |
| mean logits | $80.13 \pm 0.60$ |

(b) Similarity metrics

| Metric | Test Accuracy |
|---|---|
| cosine similarity | $\mathbf{84.05 \pm 0.53}$ |
| neg. Euclidean dist. | $81.85 \pm 0.58$ |
| unscaled dot-prod. | $37.60 \pm 0.64$ |

### D.3 Ablation regarding temperature scaling of embedding similarity logits

As reported in the main paper, we use the temperature $\tau_S$ to rescale the logits of our task-specific similarity matrix $\tilde{S}$ via division (or the original similarity matrix $S$ in case no task-specific adaptation shall be used). We investigate two different ways of temperature scaling: (i) the possibility of using a fixed temperature defined as $1/\sqrt{d}$ where $d$ is the dimension of the patch embeddings of the respective architecture, and (ii) learning the appropriate temperature during the meta fine-tuning procedure. In practice, we learn $\log(\tau_S)$ to ensure $\tau_S \geq 0$.

We observe throughout our 1-shot experiments depicted in Figure A2 (a) and (b) that the temperature converges towards our default values of $1/\sqrt{d}$ shown as a dashed horizontal line. This is independent of the initial value of the temperature parameter $\tau_S^{\text{init}}$. For the 5-way 5-shot experiments presented in Figure A2 (c) and (d) however, we observe that while our default value still achieves good results, the learned temperature converges to a slightly lower value across all experiments.

### D.4 Development over the course of pretraining

We further present insights into the development of the accuracy during self-supervised pretraining. Since our pretraining procedure is entirely unsupervised and does hence not include any labels, we investigate models trained for a variety of different epochs and evaluate these on the test set using the proposed similarity-based classification method *with* ('5 steps' and '15 steps') and *without* ('None') and present the results in Table A4. Note that no meta fine-tuning was employed here. We observe that while the performance significantly increases over the first 50 epochs, there seems to be some saturation and even slight decrease in performance until above 500 epochs where the accuracy increases again and (mostly) achieves highest results in this study.

Table A4: **Development of test accuracy in self-supervised pretraining.** Results obtained for a 5-way 5-shot scenario on the *tiered*ImageNet test set using our proposed classifier with a ViT-small backbone. For online optimisation (i.e., steps$> 0$), we use SGD with 0.1 as learning rate.

| Reweighting steps | Epochs | | | | | |
|---|---|---|---|---|---|---|
| | 1 | 50 | 100 | 250 | 500 | 800 |
| None | $39.20_{\pm0.69}$ | $73.30_{\pm0.75}$ | $73.63_{\pm0.73}$ | $72.84_{\pm0.72}$ | $71.51_{\pm0.72}$ | $73.83_{\pm0.74}$ |
| 5 steps | $39.34_{\pm0.69}$ | $73.59_{\pm0.74}$ | $74.03_{\pm0.73}$ | $73.10_{\pm0.73}$ | $71.82_{\pm0.72}$ | $74.16_{\pm0.73}$ |
| 15 steps | $39.43_{\pm0.69}$ | $73.86_{\pm0.73}$ | $74.48_{\pm0.74}$ | $73.41_{\pm0.75}$ | $72.16_{\pm0.73}$ | $74.42_{\pm0.74}$ |

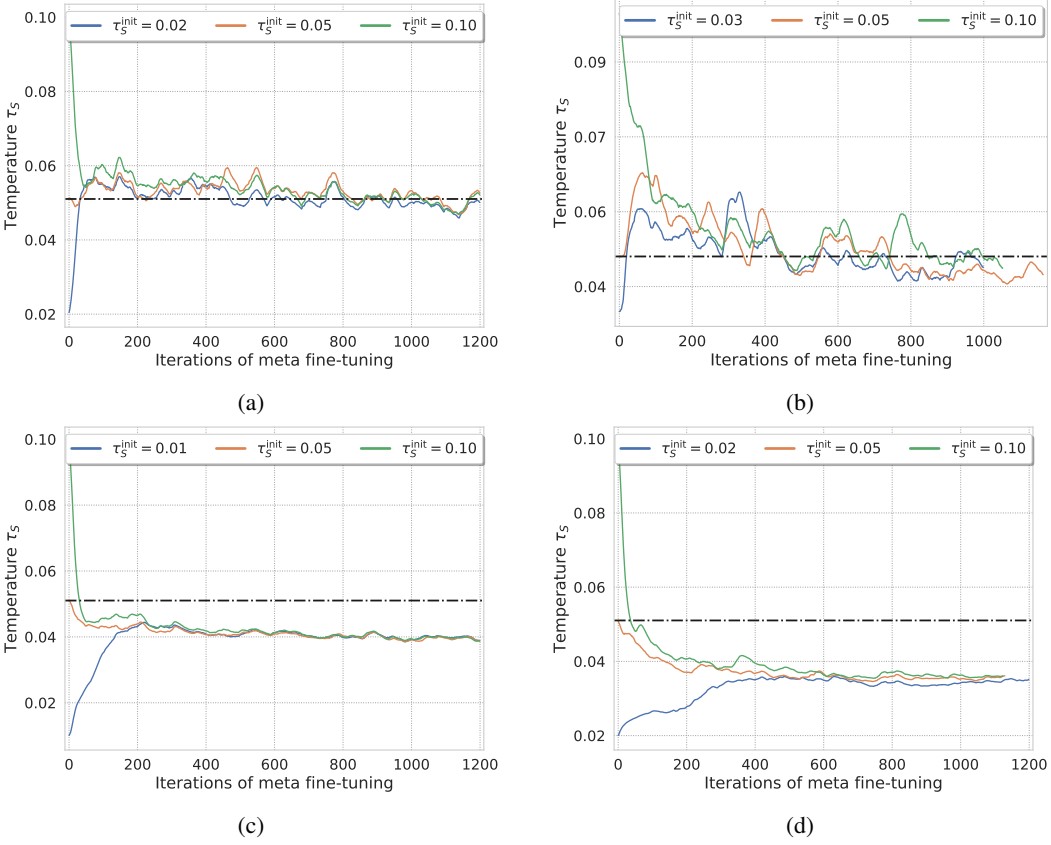

Figure A2: **Temperature for rescaling similarity logits.** (a) and (b) show the learned temperatures for 5-way 1-shot scenarios on *mini*ImageNet and *tiered*ImageNet, respectively. The corresponding 5-way 5-shot results are depicted in (c) and (d). All experiments have been conducted using a ViT-small architecture.

## E    Further visualization of instance embeddings

Figure 5 in the main paper depicts instance and class embeddings visualized via PCA projection to the three dominant dimensions. Figure A3 additionally depicts a comparison of projected views of the tokens of 5 instances from a novel class in embedding space for different ways of meta training. While the representations obtained from the network meta fine-tuned by using common averaging over the embeddings ('*average*') do not exhibit any clear separation of the instances, the embeddings obtained with our classifier seem to retain the instance information ('*w/o $v$*') and separation is improved when using token importance reweighting ('*w/ $v$*'). These results indicate that our similarity-based classifier coupled with task-specific token reweighting is able to better disentangle the embeddings of different instances from the same class, which further prevents the network from supervision collapse and helps to achieve the higher performance observed on the benchmarks.

## F    Datasets used for evaluation

We train and evaluate our approach presented in the main paper on the following few-shot image classification datasets:

***mini*ImageNet**.    The *mini*ImageNet dataset has been initially proposed by [20] with follow-up modifications by [14] and consists of a specific 100 class subset of ImageNet [16] with 600 images for each class. The data is split into 64 training, 16 validation and 20 test classes.

***tiered*ImageNet**.    Similar to the previous dataset, the *tiered*ImageNet [15] is a subset of classes selected form the bigger ImageNet [16] dataset, however with a substantially larger set of classes and

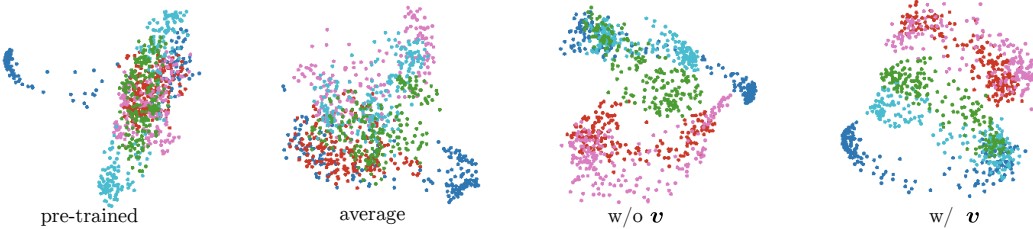

pre-trained             average             w/o $\boldsymbol{v}$             w/ $\boldsymbol{v}$

Figure A3: **Instance embeddings after meta fine-tuning.** Visualized are the projected tokens of 5 instances of the same novel support set class for different meta fine-tuning (M-FT) methods (after self-supervised pretraining). From left to right: self-supervised pretraining only, M-FT using an average embedding per class, M-FT using our classifier but without task-specific token reweighting, M-FT using our classifier with 15 reweighting steps. (Projection via PCA to main dimensions.)

different structure in mind. It comprises a selection of 34 super-classes with a total of 608 categories, totalling in 779,165 images that are split into 20,6 and 8 super-classes to achieve better separation between training, validation and testing, respectively.

**CIFAR-FS**. The CIFAR-FS dataset [1] contains the 100 categories with 600 images per category from the CIFAR100 [8] dataset which are split into 64 training, 16 validation and 20 test classes.

**FC-100**. The FC-100 dataset [12] is also derived from CIFAR100 [8] but follows a splitting strategy similar to *tiered*ImageNet to increase difficulty through higher separation, resulting in 60 training, 20 validation and 20 test classes.

# G Implementation details

We present further details regarding our implementation and used hyperparameters in the following.

## G.1 Pretraining

**GPU usage**. We pretrain our models with the use of 4 Nvidia A100 GPUs with 40GB each for our ViT [6, 19] and 8 such GPUs for our Swin [9] variants.

**Hyperparameter choice**. We follow the strategy introduced by [25] to pretrain our Transformer backbones and mostly stick to the hyperparameter settings reported in their work. We generally use two global crops and 10 local crops with crop scales of $(0.4, 1.0)$ and $(0.05, 0.4)$, respectively. We further use a patch size of 16 for our ViT models and a window size of 7 for Swin, corresponding to the default sizes for ViT-small [6, 19] and Swin-tiny [9]. We use an output dimension of 8192 for the projection heads across all models, and employ random Masked Image Modelling with prediction ratios $(0, 0.3)$ and variances $(0, 0.2)$. Our ViT and Swin architectures are trained with an image size of $224 \times 224$ arranged in batches of size 512 samples for 1600 and 800 epochs, respectively, using a linearly ramped-up learning rate (over first 10 epochs) of $5\mathrm{e}{-4} \times \mathrm{batchsize}/256$. For detailed information, we would like to refer the interested reader to the work by Zhou *et al*. [25] where more background information regarding the influence and justification of these hyperparameters is provided.

## G.2 Meta fine-tuning

**GPU usage**. During the meta fine-tuning (M-FT) stage, we use 1 and 2 Nvidia 2080-ti GPUs for ViT-small and Swin-tiny, respectively, across all 4 datasets.

**Hyperparameters**. We fix the input image size as $224 \times 224$ for all datasets. We use the SGD optimizer along with a learning rate of $2\mathrm{e}{-4}$, 0.9 as the momentum value and $5\mathrm{e}{-4}$ as the weight decay. Additionally, we employ a learning rate scheduler with cosine annealing for 5,000 iterations as one cycle, ramping down to $5\mathrm{e}{-5}$ at the end of each cycle.

**Online optimization**. During the online learning of the token importance reweighting vectors, we adopt the SGD optimizer with 0.1 as the learning rate. For online update steps, we generally choose

a default value of 15 steps across all datasets. For further details regarding the temperature scaling procedure used to rescale our task-specific similarity logits, please refer to Section D.3.