# OpenReview forum: "Rethinking Generalization in Few-Shot Classification"
_NeurIPS.cc/2022/Conference — NeurIPS 2022 Accept_

### Official Review · Reviewer_rwKM · 2022-07-10

**Rating:** 5
**Confidence:** 5
**Soundness:** 3 good
**Presentation:** 3 good
**Contribution:** 2 fair

**Summary:**

This paper adopts self-supervised trained Vision Transformer (ViT) architecture as the feature extractor, deriving patch-level representations for few-shot classification problems. To exploit the relation across patches, the authors propose a token importance reweighting mechanism (which is required to perform during both training and testing stages). The experimental results show satisfactory performance in several commonly-used FSL benchmarks to verify the effectiveness of this method.

**Questions:**

Please see my questions raised in the above weakness part.

**Limitations:**

The authors did provide discussions on the limitation of the proposed work.

**Strengths And Weaknesses:**

The overall paper is easy to follow. The idea of utilizing the self-supervised trained ViT architecture to derive patch-level representations is interesting and seems effective for few-shot learning tasks. Since self-supervised model pre-training is agnostic to the image-level class labels, the trained model is more generalizable for downstream tasks. Also, applying ViT to extract patch-level features allows the model to produce more fine-grained information. However, I have the following concerns about this work:

1. In Fig. 4, the authors show that self-supervised pre-training performs significantly better than the supervised-pre-training counterpart. However, in prior SSL literature (e.g., [16, A]), SSL pre-training only slightly outperforms supervised pre-training (sometimes or even worse than it). A proper explanation (or insight) that the proposed SSL pre-training surpasses a large margin over the supervised counterpart reported in Fig. 4 is needed.

2. The t-SNE visualization in Fig. 5, only verifies that the patch-level embeddings derived from the same “instance” are clustered together, and those from different instances are separated from each other. However, this figure only explains/visualizes separation between different instances, but not the discrimination between different classes (which is much more important for FSL). It is desirable to see whether the embeddings extracted from the same “class” are gathered, while the embedding of different classes would separate far from each other.

3. Since the title of this paper emphasizes the aspect of “generalization” in few-shot learning, one would expect learning strategies with results/comparisons with recent cross-domain few-shot learning works (e.g., [B]). In other words, cross-domain FSL aims to transfer the learned knowledge to the novel classes in unseen target domains (showing generalization ability).

4. Although this paper applies Masked Image Modeling (MIM) as the pretext task for pre-training ViT, use of other self-supervised pre-training approaches like contrastive learning (e.g., DINO [C], MoCo v3 [A]) would be possible. It will be also good if the authors provide some insights or comparisons about the choice of the self-supervised pre-training approach. If MIM is desirable for this task, more explanations and supports would be needed.

5. The need to perform additional learning for test data (few-shot instances from novel classes) is needed for the proposed work (but not necessarily for a number of SOTAs). I'd like to see how the authors would elaborate on this issue.

[16] He et al. “Masked autoencoders are scalable vision learners.” CVPR 2022
[A] Chen et al. “An Empirical Study of Training Self-Supervised Vision Transformers.” ICCV 2021
[B] Chen et al. “A Closer Look at Few-shot Classification.” ICLR 2019
[C] Caron et al. “Emerging Properties in Self-Supervised Vision Transformers.” ICCV 2021

---

> ### Author Response · Authors · 2022-08-02
> **Response to Reviewer rwKM Part 1**
>
> We thank you for the feedback, and will address your concerns in the following point by point:
>
> >_In Fig. 4, the authors show that self-supervised pre-training performs significantly better than the supervised-pre-training counterpart. However, in prior SSL literature (e.g., [16, A]), SSL pre-training only slightly outperforms supervised pre-training (sometimes or even worse than it). A proper explanation (or insight) that the proposed SSL pre-training surpasses a large margin over the supervised counterpart reported in Fig. 4 is needed._
>
> Few-shot learning is distinctively different from conventional classification (like [16,A]) in one important aspect: novel previously unseen classes are encountered at test time. As such, supervised learning induces a tendency of the representation space to overfit to the structure of the classes observed during training. In other words, the representation space is created and condensed to easily separate observed training classes, but at the expense of distorting other dimensions that might be crucial to correctly distinguish yet unseen classes. This is known in the few-shot literature as ‘supervision collapse’ [4 ]. Since no class labels are provided during the self-supervised pre-training, we expected the method to create a more general/less distorted representation space that is significantly better suited to generalize to yet unseen classes and avoid collapse. These intuitions are supported by the results we have obtained (Fig 4.). We further
> observe that self-supervised training is helpful to prevent early overfitting when learning from small few-shot datasets (e.g. 38.4K miniImageNet vs. 1.2M ImageNet1K).
>
> >_The t-SNE visualization in Fig. 5, only verifies that the patch-level embeddings derived from the same “instance” are clustered together, and those from different instances are separated from each other. However, this figure only explains/visualizes separation between different instances, but not the discrimination between different classes (which is much more important for FSL). It is desirable to see whether the embeddings extracted from the same “class” are gathered, while the embedding of different classes would separate far from each other._
>
> We have included two new PCA visualisations of the embeddings for the entire support set of a 5-way 5-shot setting into the uploaded first revised version of our paper, with different classes indicated by color. Note that we obtain one embedding/token per image patch, and we thus expect a much larger spread of the embeddings within one class as well as higher partial overlap between classes due to similarities in background and non-class-relevant objects present in the scene, compared to other (e.g. prototype-based) methods. As can be clearly observed in the rightmost sub-figure, our token importance reweighting-based method is able to determine the essential parts
> of the image that are characteristic for each class (highlighted by the respective class color) while excluding irrelevant tokens like background or non-primary objects (highlighted in brighter color). In this way, our method is able to reliably separate instances of different classes even in cases where the images contain multiple objects and depict complex real-world scenes – in other words determine what matters most in an image for the task at hand (as shown in Figure 6).
>
> >_Since the title of this paper emphasizes the aspect of “generalization” in few-shot learning, one would expect learning strategies with results/comparisons with recent cross-domain few-shot learning works (e.g., [B]). In other words, cross-domain FSL aims to transfer the learned knowledge to the novel classes in unseen target domains (showing generalization ability)._
>
> We would like to clarify that in our work, we are not using the term ‘generalization’ as an indication of recent cross-domain scenarios but rather to describe the ‘generalization’ aspect that is inherent in few-shot classification tasks: the generalization of our representations trained on data from the training distribution towards samples of novel classes from the test data distribution. As we discuss in the introductory section of our paper, using single labels for complex or multi-object scenes can easily lead to supervision collapse – meaning that the representation space overfits to the structure of classes encountered during training (as discussed above). As our results demonstrate, our approach using self-supervised pre-training followed by meta fine-tuning with our token reweighting approach is able to obtain new state-of-the-art results on several few-shot learning benchmarks by generalizing towards the test class distribution.

---

> > ### Author Response · Authors · 2022-08-02
> > **Response to Reviewer rwKM Part 2**
> >
> > ... continued from previous
> > >_Although this paper applies Masked Image Modeling (MIM) as the pretext task for pre-training ViT, use of other self-supervised pre-training approaches like contrastive learning (e.g., DINO [C], MoCo v3 [A]) would be possible. It will be also good if the authors provide some insights or comparisons about the choice of the self-supervised pre-training approach. If MIM is desirable for this task, more explanations and supports would be needed._
> >
> > Our obtained experimental results clearly indicated that enforcing an explicit loss onto the
> > patch tokens via the pretext task of Masked Image Modelling (MIM) indeed significantly helps to
> > build strong embeddings that encode the semantic content of the associated patch. We have run
> > our initial experiments using DINO which does not enforce such local constraints and analyzed
> > the representation quality of the patch tokens w.r.t. token similarity across different samples of
> > one class by visualising the top-k most-similar tokens in a side-by-side comparison between two
> > images. We observed significantly worse and less reliable matches when using DINO compared to
> > our choice with MIM as pretext task. We are happy include visualisations together with a discussion
> > of these findings into the supplementary material of the revised paper.
> >
> > >_The need to perform additional learning for test data (few-shot instances from
> > novel classes) is needed for the proposed work (but not necessarily for a number of SOTAs). I’d like to see
> > how the authors would elaborate on this issue._
> >
> > Zero-update methods have often proven successful since updating the actual model at
> > inference time on such small number of samples (5-25) easily leads to overfitting, however at the
> > downside of not being able to adapt to the task at hand. Recent methods such as CAN[18] and FEAT [52] have
> > attempted to solve the problem by learning an extra module to predict a task-specific refinement of their
> > representations – however the learner itself remains fixed at inference time and thus relies on
> > sufficiently good a-priori training. We are in contrast learning a small set of parameters on the fly at
> > inference time (scalar weights for our correspondences) once per task before classifying the query
> > samples. This allows our method to be adaptive to the structure of the class set encountered for
> > the task at hand and leverage the inter- and intra-class dependencies without requiring additional
> > learnt modules. Table A1 in the supplementary material provides insights into the efficiency vs.
> > effectiveness trade-off. As the results suggest, our method can gain significant performance
> > improvement (more than 1% within 5 update steps) without noticeably sacrificing efficiency
> > (≈ 3ms per task). Further improvement can be observed by increasing the optimization steps.

---

> > ### Comment · Reviewer_rwKM · 2022-08-08
> > **Clarification on Q1**
> >
> > I think Q1 is the only issue that remains unclear to me. I agree that supervised learning induces a tendency of the representation space to overfit to the structure of the classes observed during training (i.e., ‘supervision collapse’ problem). The authors presented a self-supervised pre-training strategy, creating a more general/less distorted representation space that is significantly better suited to generalize to yet unseen classes and avoid collapse. However, for the supervised pre-training for the same FSL task in Fig. 4, what exactly is done? (so that the comparison to the proposed scheme is fair)

---

> > > ### Author Response · Authors · 2022-08-09
> > > **Further details to address Q1**
> > >
> > > Thank you for your continued feedback!
> > >
> > > > _[...] for the supervised pre-training for the same FSL task in Fig. 4, what exactly is done?_
> > >
> > > For adequate comparison to related work in FSL, we follow the widely adopted pretraining scheme used in FEAT [52] and other works (e.g. DeepEMD [53]) for our supervised pretraining. In detail, we train the network with cross-entropy loss on the training set of the respective dataset to solve a standard classification task (e.g. for miniImageNet: 64 classes) – i.e., using the exact same data we use for self-supervised pretraining. Like [52] we use the representations of the penultimate layer (before the classifier) to evaluate the performance and quality of the embeddings. To judge suitability of the encoder for few-shot tasks, an N-way 1-shot task is commonly solved (e.g. N=16 for miniImageNet due to the 16 classes in the validation set) – and we tried three different variants here:
> > >
> > >  1.) & 2.) One sample per class is encoded to produce a class-embedding (’prototype’), and classification performance is evaluated using 15 queries per class. (This is the method used in recent related works). To retrieve one embedding per sample, we use the average over all patch tokens produced by the Transformer architecture. For fairness regarding metrics, we evaluate both:
> > > 1) _embedding distance_ (MSE) and
> > > 2) _embedding similarity_ (cosine) to perform classification.
> > >
> > > 3.) We additionally use our own patch-based classifier to evaluate the FSL setting using all patch embeddings (as we later do during fine-tuning & evaluation).
> > >
> > > We perform validation over 200 such FSL-tasks after every epoch during training and pick the best-performing model regarding highest average validation accuracy. We encountered clear signs of overfitting during this type of training, with the training accuracy
> > > consistently improving to convergence, but validation accuracy plateauing (or decreasing) rather early on (~350-500ep), independent of the variant we used to evaluate on the validation set.
> > >
> > > We thank you for pointing this out and will include a comprehensive list detailing all used hyperparameters into the supplementary material of the paper.

---

### Official Review · Reviewer_erRy · 2022-07-10

**Rating:** 7
**Confidence:** 4
**Soundness:** 4 excellent
**Presentation:** 4 excellent
**Contribution:** 3 good

**Summary:**

The paper addresses the problem of few-shot classification. The main idea is to establish semantic correspondences between the patches from the support and the query images. These correspondences are then used to reason which class a query image belongs to. In order to down-weight the impact of background patches when performing classification, the authors also introduce an online optimization strategy to determine which patches in the support images are most informative when performing few-shot classification. The method uses the Vision Transformer to encode the patches in the support and query images. In order to learn strong generic features, the Vision Transformer is trained in an unsupervised manner using the masked image modelling task. The self-supervised pre-training is shown to provide better results then the supervised counterpart. The proposed method obtains state-of-the-art results on four few-shot classification benchmarks.

**Questions:**

Please check the comments listed under Weaknesses.

**Limitations:**

The authors discuss the limitations of their work.

**Strengths And Weaknesses:**

## Strengths
**S1**: The paper is well written and easy to read.

**S2**: The proposed few-shot classifier using patch-wise correspondences is novel and interesting. The online optimization allows determining which regions are most crucial to perform classification and can be helpful specially in case of clutter in the support set images. The use of patch-wise correspondences allows determining the class of the query image by jointly reasoning over the support set as well as the query.

**S3**: The self-supervised pre-training of the Vision Transformer makes sense, specially in the context of few-shot learning, to learn generic feature representation.

**S4**: The proposed approach is shown to obtain state-of-the-art results on 4 standard benchmarks (mini ImageNet, Tiered ImageNet, CIFAR-FS, FC100).

**S5**: The authors provide helpful analysis and ablation studies, showing the impact of the major contributions.


## Weaknesses
I do not have any major issues with the paper. Some minor issues which could be addresses

**W1**: As shown in Fig. 4, the use of self-supervised pre-training for Vision Transformer provides a significant improvement, compared to supervised training. It would be interesting to see what performance existing classifiers (e.g. learning a linear classifier, prototype) obtain when using the same backbone network. This would help evaluate the benefits of proposed embedding similarity based classifier.

**W2**: In the sota comparison, most previous methods use ResNet-12 backbone, while the authors employ ViT-small and Swin-Tiny. It will be helpful if the authors include the model sizes (number of parameters) for each of these backbones for comparison.

**W3**: The approach computes patch-wise correspondence between all the support and query images. I wonder if this could become computationally expensive when dealing which large number of number classes, or when each class has more samples (e.g. 100 way - 30 shot classification). A discussion on this would be beneficial.

---

> ### Author Response · Authors · 2022-08-02
> **Response to Reviewer erRy**
>
> Thank you for your review. We will address the concerns in the following point by point:
>
> >_As shown in Fig. 4, the use of self-supervised pre-training for Vision Transformer provides a significant improvement, compared to supervised training. It would be interesting to see what performance existing classifiers (e.g. learning a linear classifier, prototype) obtain when using the same backbone network. This would help evaluate the benefits of the proposed embedding similarity-based classifier._
>
> We have run additional experiments using our pre-trained ViT-small backbone followed
> by classifier-specific meta fine-tuning to provide further insights into the performance of existing classifiers. We were able to obtain a test accuracy of 82.80% for the prototypical network after optimising the pre-trained backbone with meta-finetuning, which is competitive to the results we obtain for our method without reweighting (’0 step’ in  Fig.7 (b)) but is still clearly outperformed by our reweighting-based approach (see table below). To provide a fair comparison, we optimize the linear classifier at inference time to adapt to the support set and obtained a maximum test accuracy of 82.37%. Both results indicate the quality of embedding our backbone is able to produce but also demonstrate
> the importance of our task-specific reweighting-based approach.
>
> | Model    | Test Acc    |
> | :---        |    :----:   |
> |Protonet w/ Euclidean distance| 82.80±0.59|
> |ProtoNet w/ Cosine distance | 79.90±0.65|
> |Linear classifier | 82.37±0.57|
> |FewTURE (ours) 0 rew. steps|82.68±0.55|
> |FewTURE (ours) 15 rew. steps|**84.05±0.53**|
>
> >_In the sota comparison, most previous methods use ResNet-12 backbone, while the authors employ ViT-small and Swin-Tiny. It will be helpful if the authors include the model sizes (number of parameters) for each of these backbones for comparison._
>
> We have included the model sizes as well as some additional state-of-the-art baselines (using WRN-28-10) into our comparison to the state-of-the-art. We will further include a more in-detailed discussion regarding the influence of model size into the supplementary material.
>
> >_The approach computes patch-wise correspondence between all the support and query images. I wonder if this could become computationally expensive when dealing which large number of number classes, or when each class has more samples (e.g. 100 way - 30 shot classification). A discussion on this would be beneficial._
>
> We thank you for pointing this out and acknowledge that this is indeed an important point the needs consideration. Our method can be easily adapted by pruning the number of considered tokens and thus scaled to the large setting _i.e.,_ many-way many-shot. We have run experiments using the attention maps inherent in our approach to prune our number of patch tokens and only use the top-k for increased computational efficiency. Using our ViT-small backbone, we trained and evaluated pruning the number of tokens to 75%, 50%, 25% and 10% of the original token number and obtained the following results:
>
> | # tokens    | Test Acc    |
> | :---        |    :----:   |
> |100%| 84.05±0.53|
> | 75%| 83.15±0.57|
> | 50%|82.81±0.59|
> | 25%|81.79±0.57|
> | 10%|81.05±0.62|

---

> > ### Comment · Reviewer_erRy · 2022-08-07
> > **Thanks for the response**
> >
> > Thank you for providing detailed answers. I will stick to my original rating and recommend acceptance.

---

### Official Review · Reviewer_xXQn · 2022-07-11

**Rating:** 7
**Confidence:** 5
**Soundness:** 3 good
**Presentation:** 4 excellent
**Contribution:** 4 excellent

**Summary:**

Motivated by supervision collapse caused by standard few-shot training on weak image-level labels, authors introduce a token-based approach based on unsupervised vision transformers that reweights tokens in an inner loop based on their discriminative power. The model achieves strong results and demonstrates the viability of vision transformer models on few-shot tasks without extra pre-training.

**Questions:**

My main questions involve the two broad weaknesses outlined above.

1. How should readers interpret the difference in model backbone sizes, and how does this impact the presented results?

2. Can authors provide or elaborate on the missing ablations – what is the performance contribution of the vision transformer backbone vs token reweighting vs logsumexp aggregation? And less importantly, how significant was the choice of similarity metric?

Less impactful questions/comments:

3. Is the inner-loop token-reweighting scheme compatible with more sophisticated patch-to-patch techniques, such as CTX or FRN? Or is this (understandably) left for future work?

4. Given a limited computation budget this may not be feasible, but it would be interesting to see if the results from Fig4 and Sec3.2 hold for a more standard meta-fine-tuning technique (i.e. a basic Prototypical setup). Authors appear to have produced such a model in at least one setting for Fig5 (“average”). What does the performance look like? Can authors elaborate or speculate on this?

Some small typos:

•	Pg2 line 58: extend  extent

•	Pg2 line 66: class is in  class in

•	Pg3 footnote: in generally  is generally

•	Pg4 line 121: The in this way introduced  In this way, introduced / The introduced

•	Pg4 line 158: device  devise

•	Pg7 fig7 caption: SDG  SGD

**Limitations:**

The analysis of 1-shot effectiveness and discussion of smaller training datasets is insightful. The entanglement of vision transformer benefits with token reweighting benefits in presented results is not discussed. Societal impacts are not discussed, though do not extend beyond those of few-shot learning in general.

**Strengths And Weaknesses:**

STRENGTHS:

•	Demonstrated use of vision transformers for few-shot learning is on its own a neat contribution

•	Demonstrated use of purely unsupervised pre-training for few-shot learning is also a neat contribution, if not quite as novel

•	Results are impressive and span multiple benchmarks and architectures

•	Straightforward and sensible approach to token aggregation and reweighting

WEAKNESSES:

•	The strong results come with a major caveat: the ViT-small and Swin-tiny architectures have 22M and 29M parameters respectively, while the compared baselines are almost entirely based on ResNet12, which by my recollection has only 12M parameters. While this is still less than the widely-used WRN-28-10 backbone (36.5M params), I worry that the comparisons presented in the paper are apples-to-oranges. The difference in model size should be discussed and addressed.

•	Use of vision transformers for few shot classification deserves an empirical study all on its own. Understandably this is not provided here, but because of this it is unclear to what degree improvement is coming from the token reweighting scheme (in theory compatible with existing convolutional architectures) vs the vision transformer backbone (in theory compatible with existing few-shot classifiers). For example, how does the token reweighting scheme compare to simply training a linear classifier head on the support features from a vision transformer? Admittedly, a full comparison along both these axes would be clearly out of scope here.

•	Similarly, there is no ablation study provided for the impact of token reweighting vs the logsumexp aggregation scheme. How much better is logsumexp aggregation than direct addition, for example, which would correspond to basic prototype comparison with reweighted averages on each prototype? More broadly, it appears that the token reweighting scheme is broadly compatible with many existing token-to-token classifiers such as CTX and FRN, and it is not clear how the logsumexp aggregator compares.

•	More generally, the approach, while straightforward and sensible, does contain a few design choices that are not fully explained or empirically justified (for example, in addition to above, the choice of token similarity metric).

•	A slightly relevant omitted citation: the masked inner token reweighting scheme might possibly owe some conceptual debt to Batch Folding from [Few Shot Learning with Localization in Realistic Settings, CVPR2019], which also models a support-to-support classification task with an identical image-masked leave-one-out scheme (though admittedly implemented quite differently).

---

> ### Author Response · Authors · 2022-08-02
> **Response to Reviewer xXQn Part1**
>
> We thank you for your detailed feedback, which we will address in the following point by point:
>
> >The strong results come with a major caveat: [...] The difference in model size should be discussed and addressed.
>
> >How should readers interpret the difference in model backbone sizes, and how does this impact the presented results?
>
> We thank you for drawing our attention to this aspect of our work, and would like to address this in three ways:
> 1. We included the most recently published works (2021 & 2022) to ensure fair comparison, the majority of which use the ResNet12 backbone. While many other popular works indeed use WRN-28-10 (_e.g._ S2M2 [31], LEO [38], CC [15]), most have been outperformed by the more recent ResNet12-based methods. We have additionally added the two recent WRN methods OM [35] and PSST [8].
> We are happy to include a more extensive comparison to the state-of-the-art (including previous years) into the supplementary material.
>
> 2. Related works have shown that model size seems to not be a good indicator for few-shot performance, most likely since training datasets are comparably small (_e.g._ 38.4K images in miniImageNet vs. standard ImageNet with 1.28M) and big networks are thus much more prone to overfit.
> Chen _et al._ [Chen] demonstrate in Figure 3 of their paper that the performance gains due to larger backbones plateau across all methods for backbones bigger than ResNet10 and only offer diminishing gains (if any at all). The investigations of Mangla _et al._[31] yielded similar results, where the performance on the miniImageNet and tieredImageNet datasets even decreased by around 0.5-1% when scaling up from ResNet18 to ResNet34 (Table 2).
> We thus conclude that increased number of parameters on its own does not lead to better few-shot performance, and the tendency of many recent works to choose the established ResNet12 (12.4M) over bigger backbones is highly likely a result of this.
> We will add additional discussion regarding the model size and its potential influence to the supplementary material of our paper (due to space limitations of the main paper's body).
>
> 3. We have run additional experiments using the significantly smaller ViT-tiny architecture with only 5M parameters [45]. Initial results show that our method achieves a competitive accuracy of 81.10% on the miniImageNet test dataset with less than one seventh of the number of parameters of WRN-28-10:
>
> |Model | Backbone |Params| Test Acc.|
> |:---|:----:|---:|---:|
> |OM [35]| WRN28-10| ≈ 36.5M|85.29±0.4|
> |FewTURE (ours)|ViT-Tiny| ≈ 5M|81.10±0.61|
> |FewTURE (ours)|ViT-Small| ≈ 22M|84.51±0.53|
> |FewTURE (ours)|Swin-Tiny| ≈ 29M|**86.38±0.49**|
>
> [Chen] Chen, Wei-Yu, et al. "A closer look at few-shot classification.", ICLR 2019
>
> > [...] missing ablations – what is the performance contribution of the vision transformer backbone vs token reweighting vs logsumexp aggregation? And less importantly, how significant was the choice of similarity metric?
>
> 1. Token reweighting:  The contribution of our token reweighting scheme to the model’s performance can be seen in Figure 7 (b) of our paper, where we show the achieved test accuracies for our method trained and evaluated with different numbers of reweighting
> steps. As can be seen, ’5 steps’ show a significant improvement in performance (82.68% to 83.83%) over the ’0 step’ variant (no reweighting) – demonstrating the importance of this component.
>
> 2. Aggregation: We use the logsumexp operation for our aggregation as it poses a rigorous and numerically stable way of combining individual class probabilities (one for each token) to a valid overall probability distribution over classes for each image, independent of
> how the individual token probability scores are obtained (see similarity metrics below). We have run additional experiments (training and testing) using our method (ViT-small) and 15 token reweighting steps with the only change being the suggested aggregation via sum
> (mean), and found it to underperform our proposed logsumexp method of aggregation. Direct addition without normalization (mean) proved highly unstable due to large logit values.
> |Method|Test Acc.|
> |:---|:----:|
> |mean logits|80.13±0.60|
> |logsumexp| **84.05±0.53**|
>
> 3. Similarity metrics. We have investigated the use of negative mean Euclidean distance and unscaled dot-product as alternative, and found our proposed cosine similarity to outperform both:
> |Metric|Test Acc.|
> |:---|:----:|
> |Cosine|**84.05±0.53**|
> |neg MSE|81.85±0.58|
> |Cosine|37.60±0.64|
>
> We thank you for pointing out these ablations and will include the additional results with discussion into the supplementary material of our revised paper.

---

> > ### Author Response · Authors · 2022-08-02
> > **Response to Reviewer xXQn Part2**
> >
> > _... continued from previous_
> >
> > >_Is the inner-loop token-reweighting scheme compatible with more sophisticated patch-to-patch techniques, such as CTX or FRN? Or is this (understandably) left for future work?_
> >
> > Thanks for pointing out these two works in this context. CTX [9] proposes to reweight support features and produce query-aligned prototypes via using a Transformer-styled cross attention mechanism between a class of support features and the query feature, while FRN [48] proposes to reconstruct the query features via a weighted-sum of a pool of support class features. The weighting mechanisms of both methods are based on the query features and performed to refine/create embeddings. In contrast, our task-specific reweighting mechanism depends on the support set and its labels and works directly on the embedding similarity matrix, which makes it distinctly different from FRN and CTX. Developing a similarity-based reweigthing mechanism that additionally leverages query features could however be an interesting future work.
> >
> > >_Given a limited computation budget this may not be feasible, but it would be interesting to see if the results
> > from Fig4 and Sec3.2 hold for a more standard meta-fine-tuning technique (i.e. a basic Prototypical setup).
> > Authors appear to have produced such a model in at least one setting for Fig5 (“average”). What does the
> > performance look like? Can authors elaborate or speculate on this?_
> >
> > We have run additional meta fine-tuning experiments and trained both a linear classifier as well as prototypical approach using our pre-trained ViT-small backbone for a 5-way 5-shot setting on miniImagenet, obtaining the results shown in the table below. While both achieve
> > competitive performance, both are clearly outperformed by our proposed method. We optimized the linear classifier at inference time to allow sufficient adaptation to the task at hand and provide fair comparison. It is worth nothing that while the linear classifier as well as our method can efficiently adapt to the task at inference time, the prototypical does not offer this capability.
> >
> > | Model   |  Test Acc |
> > |----------|:-------------:|
> > | Protonet w/ Euclidean distance | 82.80±0.59 |
> > | ProtoNet w/ Cosine distance | 79.90±0.65 |
> > | Linear classifier |  82.37±0.57 |
> > | FewTURE (ours) 0 rew. steps |    82.68±0.55 |
> > | FewTURE (ours) 15 rew. steps |  **84.05±0.53** |
> >
> > > _A slightly relevant omitted citation: the masked inner token reweighting scheme
> > might possibly owe some conceptual debt to Batch Folding from [B][Few Shot Learning with Localization in
> > Realistic Settings, CVPR2019], which also models a support-to-support classification task with an identical
> > image-masked leave-one-out scheme (though admittedly implemented quite differently)_
> >
> > We thank you for pointing out this work’s relation to our method and will include it into our next revision. While we see the general idea of image-masked ’leave-one-out’ scheme as a possible similarity, the implementation is (as indicated) significantly different between both methods. We use image-wise (5-shot) or token-wise (1-shot) masking of patch-similarities while still strictly adhering to the few-shot learning split of support and query set, and only use the actual support images and labels in our adaptation strategy whereas [B] employs a leave-one-out cross-validation scheme across all images the entire batch (support and query images) to increase the number of data samples, reduce gradient noise and learn better representations.
> >
> > > _[...] small typos_
> >
> > Thank you for pointing these out, we have corrected them in our revised version.

---

> > > ### Comment · Reviewer_xXQn · 2022-08-03
> > > **Thank you for the response, score updated**
> > >
> > > Thank you for the detailed response, the new results are quite interesting and my concerns (shared with Reviewer erRy) are for the most part addressed. Apologies for missing the implication in Fig.7b. Regarding token reweighting for CTX/FRN, I had been envisioning a setup with batch folding, for a similarly support-set-based training scheme, but batch folding is in no way universal and yes, such an adaptation would clearly be out of scope. Regarding batch folding, I’m not sure I understand the distinction between training with a support-query split, and using only the supports, versus training with a smaller batch and all images as supports, but this is ultimately a semantic argument and not particularly relevant to my scoring of the paper at this point. Final rating raised to Accept.

---

### Official Review · Reviewer_eF5W · 2022-07-13

**Rating:** 6
**Confidence:** 3
**Soundness:** 3 good
**Presentation:** 3 good
**Contribution:** 3 good

**Summary:**

This works deal with the image-level annotation problem in few-shot classification. During training time, given support and query images, the authors first split images into patches, use a transformer to extract feature tokens, and learn which tokens are related to the label more. During test time, it self-learned within support images to find which tokens are more important. The experiment shows a consistent improvement.

**Questions:**

It is unclear to me what "meta fine-tuning" means. Does it mean the step of fine-tuning the model with support data at interference time?


**Limitations:**

Yes, the author addressed the potential problem when training data is highly limited.

**Strengths And Weaknesses:**

Strength:
The transformer-only architecture makes it very clean and thus makes it potentially be a new baseline of the problem. Reweighting features to focus on task-specific information is also a reasonable idea. The experiment shows a consistent improvement.

Weakness:
It is better to discuss other feature re-weighting-based methods in few-shot classification. For example, for fine-grained few-shot classification, Lee et al. [A] shows a consistent improvement in all previous methods by adding an attention-based feature re-weighting module.

[A] Lee et al., "Task Discrepancy Maximization for Fine-Grained Few-Shot Classification.", CVPR 2022.

---

> ### Author Response · Authors · 2022-08-02
> **Response to Reviewer eF5W**
>
> We thank you for your review. We will address your feedback in the following point by point:
>
> >_It is better to discuss other feature re-weighting-based methods in few-shot classification. For example, for fine-grained few-shot classification, Lee et al. [A] shows a consistent improvement in all previous methods by adding an attention-based feature re-weighting module._
>
> We thank you for pointing out this interesting work. While we discuss other related methods that can be interpreted as spatial (_e.g._ CTX [9], CAN [18]) or channel-wise feature reweighting (_e.g._ FEAT [52]), this very recent work has not yet been included.
> In contrast to previous works, both TDM [A] and our method share the idea of using the entire support set to determine helpful inter- and intra-class information to solve the task, however differ significantly in the way how the challenge is approached.
> [A] uses two attention modules to predict class- and task-specific weight vectors and transforms the feature maps extracted via a CNN backbone by reweighting the channels. In contrast, our reweighting approach is not modifying the embeddings but directly uses the similarity between patch tokens encoded by our Vision Transformer and determines a single scalar importance weight for each, _i.e._ learns which spatial regions matter.
> We will include and discuss this work in the next revision of our paper.
>
> >_It is unclear to me what "meta fine-tuning" means._
>
> We would like to apologize if our introduction of the term _'meta fine-tuning'_ in Section 3.1 has not been explicit enough and might thus have led to increased difficulty in understanding our work.
> In our paper, we use the expression _'meta fine-tuning'_ to indicate the meta training phase that follows after the initial self-supervised pretraining (_i.e._ fine-tuning of the weights). As the name indicates, this training procedure is conducted in a bi-level meta-learning manner where two cascaded loops are used (_cf._ [47], [13]). While the inner loop performs the task adaptation via our token importance weights using the provided support sets (Section 2.4), the outer loop computes the loss by evaluating the classification performance on the unseen query samples (Section 2.3), and uses this to update the parameters of the network.

---

> > ### Comment · Reviewer_eF5W · 2022-08-07
> > **Response to comment**
> >
> > Thanks to the author for the comments to address my concerns. I will still recommend a weak acceptance of this work.

---

### Meta-Review · Area_Chair_XDDy · 2022-09-03

**Recommendation:** Accept
**Confidence:** Certain

**Metareview:**


 This paper tackles few-shot learning with a transformer architecture and, inspired by the intuition that fine-grained information is ignored in existing methods, uses an inner-loop token re-weighting method to improve results. Overall the reviewers appreciated the use of modern architectures (Vision Transformers), the reasonableness of the re-weighting intuition, and experimental results. Concerns were raised about comparison to existing methods with similar intuitions (e.g. [A] mentioned by eF5W), fairness of the comparison with respect to model capacity and in general ablations demonstrating that it's the method (not transformers by themselves) leading to improved results, and lack of principled explanations for the design choices, and computational complexity.

  The authors provided strong rebuttals, including new experiments using linear classifiers and prototypical approaches, use of smaller models, and a demonstration of potential pruning methods to address computational complexity. The reviewers were overall receptive to the rebuttal, and all recommended acceptance of this paper after some back-and-forth. The paper provides both a nice benchmark applying Vision Transformers to few-shot learning as well as a method that is demonstrably better through ablation studies. Therefore, this paper provides several nice contributions to the community, and I recommend acceptance.

**Award:**

No

---

### Decision · Program_Chairs · 2022-09-14

Accept